# Adaptive coding for dynamic sensory inference

Wiktor F Młynarski[1]*, Ann M Hermundstad[2]*

[1]Department of Brain and Cognitive Sciences, Massachusetts Institute of Technology, Cambridge, United States; [2]Janelia Research Campus, Howard Hughes Medical Institute, Ashburn, United States

**Abstract** Behavior relies on the ability of sensory systems to infer properties of the environment from incoming stimuli. The accuracy of inference depends on the fidelity with which behaviorally relevant properties of stimuli are encoded in neural responses. High-fidelity encodings can be metabolically costly, but low-fidelity encodings can cause errors in inference. Here, we discuss general principles that underlie the tradeoff between encoding cost and inference error. We then derive adaptive encoding schemes that dynamically navigate this tradeoff. These optimal encodings tend to increase the fidelity of the neural representation following a change in the stimulus distribution, and reduce fidelity for stimuli that originate from a known distribution. We predict dynamical signatures of such encoding schemes and demonstrate how known phenomena, such as burst coding and firing rate adaptation, can be understood as hallmarks of optimal coding for accurate inference.

DOI: https://doi.org/10.7554/eLife.32055.001

**\*For correspondence:**
mlynar@mit.edu (WFM);
hermundstada@janelia.hhmi.org
(AMH)

**Competing interests:** The authors declare that no competing interests exist.

## Introduction

Biological systems must make inferences about the environment in order to successfully plan and accomplish goals. Inference is the process of estimating behaviorally relevant properties of the environment from low-level sensory signals registered by neurons in the early sensory periphery (*Kersten and Schrater, 2002*). Many perceptual tasks, such as color perception (*Brainard et al., 2006*), visual speed estimation (*Weiss et al., 2002*), or sound localization (*Fischer and Peña, 2011*; *Młynarski, 2015*), can be understood as probabilistic inference. All these tasks rely on the estimation of features (such as the speed of an object) that are not explicitly represented by low-level sensory stimuli (such as light signals incident on photoreceptors).

To accurately perform inference, the nervous system can construct an internal model that relates incoming sensory stimuli to behaviorally relevant properties of the environment (*Kersten and Schrater, 2002*; *Kersten et al., 2004*; *Fiser et al., 2010*; *Rao et al., 2002*; *Coen-Cagli et al., 2015*). As the environment changes, this internal model must be continually updated with new stimuli (*Wark et al., 2009*; *DeWeese and Zador, 1998*; *Nassar et al., 2010*; *Lochmann et al., 2012*; *Deneve, 2008*), and therefore the accuracy of this internal model depends on the fidelity with which incoming stimuli are encoded in neural responses.

The process of encoding sensory stimuli, however, is metabolically expensive (*Laughlin et al., 1998*; *Mehta and Schwab, 2012*; *Balasubramanian et al., 2001*; *Harris et al., 2012*; *Attwell and Laughlin, 2001*; *Levy and Baxter, 1996*), and a large body of evidence suggests that sensory systems have evolved to reduce the energetic costs of stimulus coding (*Laughlin et al., 1998*; *Laughlin and Sejnowski, 2003*; *Hermundstad et al., 2014*). These findings provide empirical support for the efficient coding hypothesis (*Barlow, 1961*), which postulates that sensory systems minimize metabolic cost while maximizing the amount of information that is encoded about a stimulus (*van Hateren, 1992*; *Olshausen and Field, 1996*; *Laughlin, 1981*).

The goal of maximizing stimulus information does not reflect the fact that different stimuli can have different utility to a system for making inferences about the environment (*Tishby et al., 2000*; *Palmer et al., 2015*; *Geisler et al., 2009*; *Burge and Geisler, 2015*). The relative utility of a stimulus is determined by the potential impact that it can have on the system's belief about the state of the environment; stimuli that sway this belief carry high utility, while stimuli that do not affect this belief are less relevant. Moreover, physically different stimuli can exert the same influence the observer's belief and can therefore be encoded in the same neural activity pattern without affecting the inference process. Such an encoding strategy decreases the fidelity of the neural representation by using the same activity pattern to represent many stimuli, and consequently reduces the amount of metabolic resources required to perform inference.

When the distribution of stimuli changes in time, as in any natural environment, both the belief about the environment (*DeWeese and Zador, 1998*) and the relative impact of different stimuli on this belief also change in time. Any system that must perform accurate inference with minimal energy must therefore *dynamically* balance the cost of encoding stimuli with the error that this encoding can introduce in the inference process. While studies have separately shown that sensory neurons dynamically adapt to changing stimulus distributions in manners that reflect either optimal encoding (*Fairhall et al., 2001*) or inference (*Wark et al., 2009*), the interplay between these two objectives is not understood.

In this work, we develop a general framework for relating low-level sensory encoding schemes to the higher level processing that ultimately supports behavior. We use this framework to explore the dynamic interplay between efficient encoding, which serves to represent the stimulus with minimal metabolic cost, and accurate inference, which serves to estimate behaviorally-relevant properties of the stimulus with minimal error. To illustrate the implications of this framework, we consider three neurally plausible encoding schemes in a simple model environment. Each encoding scheme reflects a different limitation on the representational capacity of neural responses, and consequently each represents a different strategy for reducing metabolic costs. We then generalize this framework to a visual inference task with natural stimuli.

We find that encoding schemes optimized for inference differ significantly from encoding schemes that are designed to accurately reconstruct all details of the stimulus. The latter produce neural responses that are more metabolically costly, and the resulting inference process exhibits qualitatively different inaccuracies.

Together, these results predict dynamical signatures of encoding strategies that are designed to support accurate inference, and differentiate these strategies from those that are designed to reconstruct the stimulus itself. These dynamical signatures provide a new interpretation of experimentally observed phenomena such as burst coding and firing-rate adaptation, which we argue could arise as a consequence of a dynamic tradeoff between coding cost and inference error.

## Results

### A general framework for dynamically balancing coding cost and inference error

Sensory systems use internal representations of external stimuli to build and update models of the environment. As an illustrative example, consider the task of avoiding a predator (*Figure 1A*, left column). The predator is signaled by sensory stimuli, such as patterns of light intensity or chemical odorants, that change over time. To avoid a predator, an organism must first determine whether a predator is present, and if so, which direction the predator is moving, and how fast. This inference process requires that incoming stimuli first be encoded in the spiking activity of sensory neurons. This activity must then be transmitted to downstream neurons that infer the position and speed of the predator.

Not all stimuli will be equally useful for this task, and the relative utility of different stimuli could change over time. When first trying to determine whether a predator is present, it might be crucial to encode stimulus details that could discriminate fur from grass. Once a predator has been detected, however, the details of the predator's fur are not relevant for determining its position and speed. If encoding stimuli is metabolically costly, energy should be devoted to encoding those details of the stimulus that are most useful for inferring the quantity at hand.

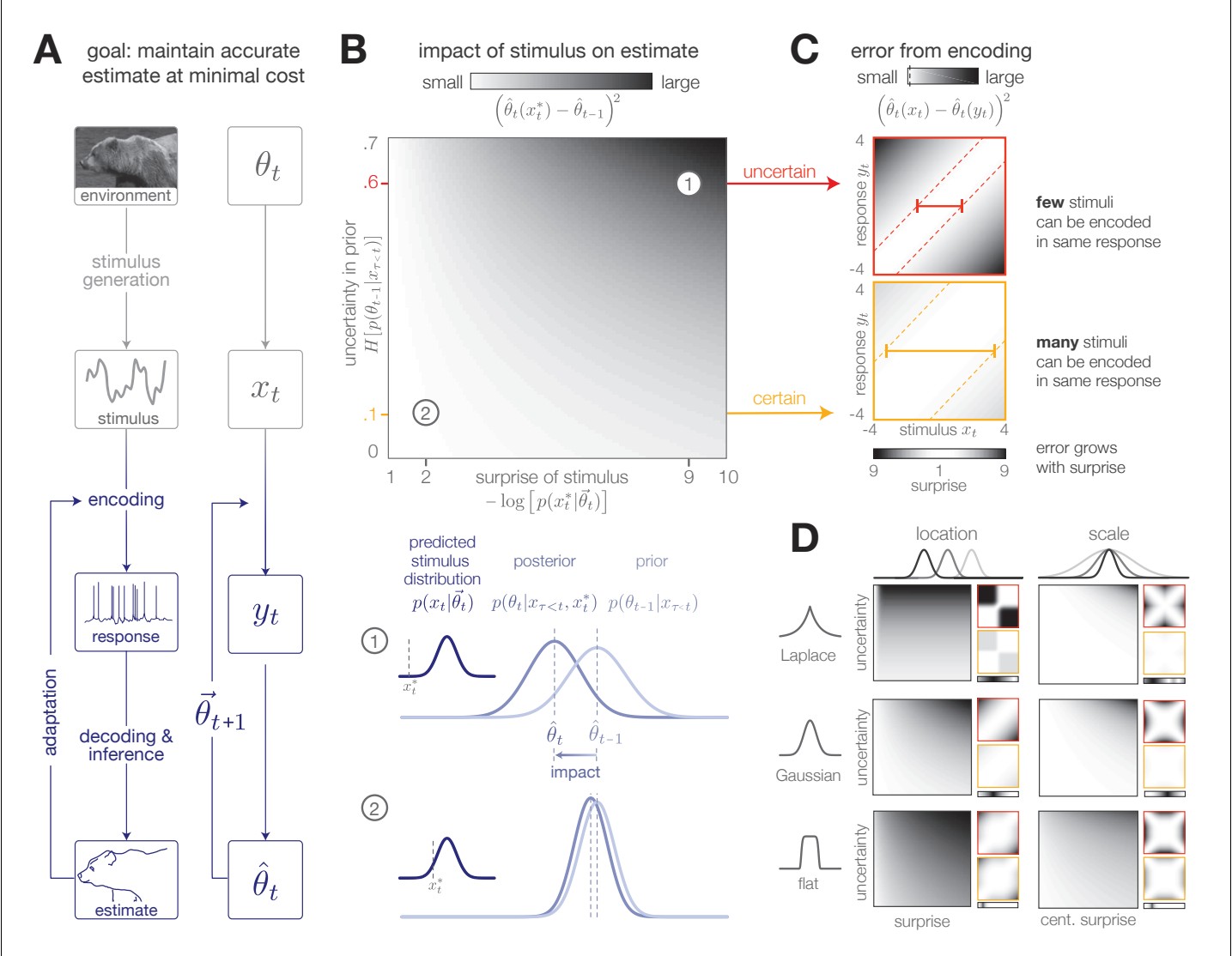

**Figure 1.** Surprise and uncertainty determine the impact of incoming stimuli for efficient inference. (**A**) We consider a framework in which a sensory system infers the state of a dynamic environment at minimal metabolic cost. The state of the environment ($\theta_t$) is signaled by sensory stimuli ($x_t$) that are encoded in neural responses ($y_t$). To infer this state, the system must decode stimuli from neural responses and use them to update an internal model of the environment (consisting of an estimate $\hat{\theta}_t$ and a prediction $\vec{\theta}_{t+1}$). This internal model can then be used to adapt the encoding at earlier stages. (The image of the bear was taken from the Berkeley Segmentation Dataset, **Martin et al., 2001**). (**B**) Incoming stimuli can have varying impact on the observer's estimate of the environmental state depending on the relationship between the observer's uncertainty and the surprise of the stimulus (heatmap). We use the example of Bayesian estimation of the mean of a stationary Gaussian distribution (**Murphy, 2007**) to demonstrate that when the observer is uncertain (wide prior $p(\theta_{t-1}|x_{\tau<t})$) and the stimulus $x_t^*$ is surprising ($x_t^*$ falls on the edge of the distribution $p(x_t|\vec{\theta}_t)$), the stimulus has high impact and causes a large shift in the posterior $p(\theta_t|x_t^*, x_{\tau<t})$ (schematic (1)). In contrast, when the observer is certain and the stimulus is expected, the stimulus has a small impact on the observer's estimate (schematic (2)). We quantify impact by the squared difference $\left(\hat{\theta}(x_t^*) - \hat{\theta}_{t-1}\right)^2$ between the estimate before and after incorporating the stimulus (Materials and methods). (Computed using $\hat{\theta}_{t-1} = 0$, for which impact spans the interval [0,0.7]). (**C**) When the observer is certain, a large number of stimuli can be mapped onto the same neural response without inducing error into the observer's estimate (orange panel). When the observer is uncertain, the same mapping from stimulus to response induces higher error (red panel). Error is highest when mapping a surprising stimulus onto an expected neural response, or vice versa. We quantify error by the squared difference $\left(\hat{\theta}_t(x_t) - \hat{\theta}_t(y_t)\right)^2$ between the estimate constructed with the stimulus versus the response (Materials and methods). Shown for uncertainty values of 0.1 (orange) and 0.6 (red). Pairs of colored dotted lines superimposed on the heatmap indicate contours of constant error tolerance $E_{tol} = 0.05$ (whose value is also marked by the vertical dotted line in the colorbar). Colored horizontal bars indicate the set of stimuli $\{x_t\}$ that can be mapped to the same neural response $y_t = 0$ with an error less than $E_{tol}$. (**D**) Qualitatively similar results to those shown in panels B-C are observed for estimating the location and scale of a

*Figure 1 continued on next page*

*Figure 1 continued*

stationary generalized Gaussian distribution. Stimuli have a larger impact on the observer's estimate when the observer is uncertain and when stimuli are unexpected (quantified by surprise in the case of location estimation, and centered surprise in the case of scale estimation; see main text). The error induced by mapping a stimulus onto a response grows with the surprise of the stimulus. For the case of scale estimation, this error is symmetric to exchanging $+x$ and $-x$, because positive and negative deviations from the mean (taken here to be 0) exert similar influence on the estimation of scale. Results are computed using $\hat{\theta}_{t-1} = 0$ (location) and $\hat{\theta}_{t-1} = 1$ (scale) and are displayed over the same ranges of uncertainty ([0,0.7]), surprise/centered surprise (*Yu et al., 2015*; *Roddey et al., 2000*), and stimulus/response ([−4,4]) as in panels B-C. Heatmaps of impact are individually scaled for each stimulus distribution relative to their minimum and maximum values; heatmaps of encoding error are scaled relative to the minimum and maximum error across both uncertainty values for a given stimulus distribution. See *Figure 1—figure supplement 2* for numerical values of color scale.

DOI: https://doi.org/10.7554/eLife.32055.002

The following figure supplements are available for figure 1:

**Figure supplement 1.** Algorithm for performing Bayesian inference with adaptively encoded stimuli.

DOI: https://doi.org/10.7554/eLife.32055.003

**Figure supplement 2.** Minimum and maximum values of the color ranges shown in *Figure 1B–D*.

DOI: https://doi.org/10.7554/eLife.32055.004

We formalize this scenario within a general Bayesian framework that consists of three components: (*i*) an environment, which is parameterized by a latent state $\theta_t$ that specifies the distribution $p(x_t|\theta_t)$ of incoming sensory stimuli $x_t$, (*ii*) an adaptive encoder, which maps incoming stimuli $x_t$ onto neural responses $y_t$, and (*iii*) an observer, which uses these neural responses to update an internal belief about the current and future states of the environment. This belief is summarized by the posterior distribution $p(\theta_t|y_{\tau \leq t})$ and is constructed by first decoding the stimulus from the neural response, and then combining the decoded stimulus with the prior belief $p(\theta_{t-1}|y_{\tau<t})$ and knowledge of environment dynamics. A prediction about the future state of the environment can be computed in an analogous manner by combining the posterior distribution with knowledge of environment dynamics (Materials and methods, *Figure 1—figure supplement 1*). This prediction is then fed back upstream and used to adapt the encoder.

In order to optimize and assess the dynamics of the system, we use the point values $\hat{\theta}_t$ and $\vec{\theta}_{t+1}$ as an estimate of the current state and prediction of the future state, respectively. The optimal point estimate is computed by averaging the posterior and is guaranteed to minimize the mean squared error between the estimated state $\hat{\theta}_t$ and the true state $\theta_t$, regardless of the form of the posterior distribution (*Robert, 2007*).

In stationary environments with fixed statistics, incoming stimuli can have varying impact on the observer's belief about the state of the environment, depending on the uncertainty in the observer's belief (measured by the entropy of the prior distribution, $H[p(\theta_{t-1}|x_{\tau<t})]$), and on the surprise of a stimulus given this belief (measured by the negative log probability of the stimulus given the current prediction, $-\log\left[p\left(x_t|\vec{\theta}_t\right)\right]$). We quantify the impact of a single stimulus $x_t^*$ by measuring the mean squared difference between the observer's estimate before and after observing the stimulus: $\left(\hat{\theta}_t(x_t^*) - \hat{\theta}_{t-1}\right)^2$. When the observer is certain about the state of the environment or when a stimulus is consistent with the observer's belief, the stimulus has little impact on the observer's belief (*Figure 1B*, illustrated for mean estimation of a stationary Gaussian distribution). Conversely, when the observer is uncertain or when the new observation is surprising, the stimulus has a large impact.

The process of encoding stimuli in neural responses can introduce additional error in the observer's estimate. Some mappings from stimuli onto responses will not alter the observer's estimate, while other mappings can significantly distort this estimate. We measure the error induced by encoding a stimulus $x_t$ in a response $y_t$ using the mean squared difference between the estimates constructed with each input: $\left(\hat{\theta}_t(x_t) - \hat{\theta}_t(y_t)\right)^2$. At times when the observer is certain, it is possible to encode many different stimuli in the same neural response without affecting the observer's estimate. However, when the observer is uncertain, some encodings can induce high error, particularly when mapping a surprising stimulus onto an expected neural response, or vice versa. These neural responses can in turn have varying impact on the observer's belief about the state of the environment.

The qualitative features of this relationship between surprise, uncertainty, and the dynamics of inference hold across a range of stimulus distributions and estimation tasks (*Figure 1D*). The specific geometry of this relationship depends on the underlying stimulus distribution and the estimated parameter. In some scenarios, surprise alone is not sufficient for determining the utility of a stimulus. For example, when the goal is to infer the spread of a distribution with a fixed mean, a decrease in spread would generate stimuli that are closer to the mean and therefore less surprising than expected. In this case, a simple function of surprise can be used to assess when stimuli are more or less surprising than predicted: $\left| H\left[ p\left( x_t | \vec{\theta_t} \right) \right] + \log\left[ p\left( x_t | \vec{\theta_t} \right) \right] \right|$, where $H\left[ p\left( x_t | \vec{\theta_t} \right) \right]$ is the entropy, or average surprise, of the predicted stimulus distribution. We refer to this as *centered surprise*, which is closely related to the information-theoretic notion of typicality (*Cover and Thomas, 2012*).

Together, the relative impact of different stimuli and the error induced by mapping stimuli onto neural responses shape the dynamics of inference. In what follows, we extend this intuition to non-stationary environments, where we show that encoding schemes that are optimized to balance coding cost and inference error exploit these relationships to devote higher coding fidelity at times when the observer is uncertain and stimuli are surprising.

## Adaptive coding for inference in nonstationary environments

To make our considerations concrete, we model an optimal Bayesian observer in a two-state environment (*Figure 2A*). Despite its simplicity, this model has been used to study the dynamics of inference in neural and perceptual systems and can generate a range of complex behaviors (*DeWeese and Zador, 1998*; *Wilson et al., 2013*; *Nassar et al., 2010*; *Radillo et al., 2017*; *Veliz-Cuba et al., 2016*). Within this model, the state variable $\theta_t$ switches randomly between a 'low' state ($\theta = \theta^L$) and a 'high' state ($\theta = \theta^H$) at a small but fixed hazard rate $h$ (we use $h = 0.01$). We take $\theta_t$ to specify either the mean or the standard deviation of a Gaussian stimulus distribution, and we refer to these as 'mean-switching' and 'variance-switching' environments, respectively. At each point in time, a single stimulus sample $x_t$ is drawn randomly from this distribution. This stimulus is encoded in a neural response and used to update the observer's belief about the environment. For a two-state environment, this belief is fully specified by the posterior probability $P_t^L$ that the environment is in the low state at time $t$. The predicted distribution of environmental states can be computed based on the probability that the environment will switch states in the next timestep: $P_{t+1}^L = P_t^L(1 - h) + \left(1 - P_t^L\right)h$. The posterior can then be used to construct a point estimate of the environmental state at time $t$: $\hat{\theta}_t = P_t^L \theta^L + \left(1 - P_t^L\right)\theta^H$ (the point prediction $\vec{\theta}_{t+1}$ can be constructed from the predicted distribution $P_{t+1}^L$ in an analogous manner). For small hazard rates (as considered here), the predicted distribution of environmental states is very close to the current posterior, and thus the prediction $\vec{\theta}_{t+1}$ can be approximated by the current estimate $\hat{\theta}_t$. Note that although the environmental states are discrete, the posterior distributions, and the point estimates constructed from them, are continuous (Materials and methods).

We consider three neurally plausible encoding schemes that reflect limitations in representational capacity. In one scheme, the encoder is constrained in the total number of distinct responses it can produce at a given time, and uses a discrete set of neural response levels to represent a stimulus ('discretization'; *Figure 2B–D*). In second scheme, the encoder is constrained in dynamic range and temporal acuity, and filters incoming stimuli in time ('temporal filtering'; *Figure 2E–G*). Finally, we consider an encoder that is constrained in the total amount of activity that can be used to encode a stimulus, and must therefore selectively encode certain stimuli and not others ('stimulus selection'; *Figure 2H–J*). For each scheme, we impose a global constraint that controls the maximum fidelity of the encoding. We then adapt the instantaneous fidelity of the encoding subject to this global constraint. We do so by choosing the parameters of the encoding to minimize the *error in inference*, $\left(\hat{\theta}_t(x_t) - \hat{\theta}_t(y_t)\right)^2$, when averaged over the predicted distribution of stimuli, $p\left(x_t | \vec{\theta}_t\right)$. (In what follows, we will use $\hat{\theta}_t$ and $\vec{\theta}_{t+1}$ to denote the estimates and predictions constructed from the neural response $y_t$. When differentiating between $\hat{\theta}_t(x_t)$ and $\hat{\theta}_t(y_t)$, we will use the shorthand notation $\hat{\theta}_{x,t}$ and $\hat{\theta}_{y,t}$, respectively). We compare this minimization to one in which the goal is to reconstruct the stimulus itself; in this case, the *error in reconstruction* is given by $(x_t - y_t)^2$. In both cases, the goal of minimizing error (in either inference or reconstruction) is balanced with the goal of minimizing metabolic

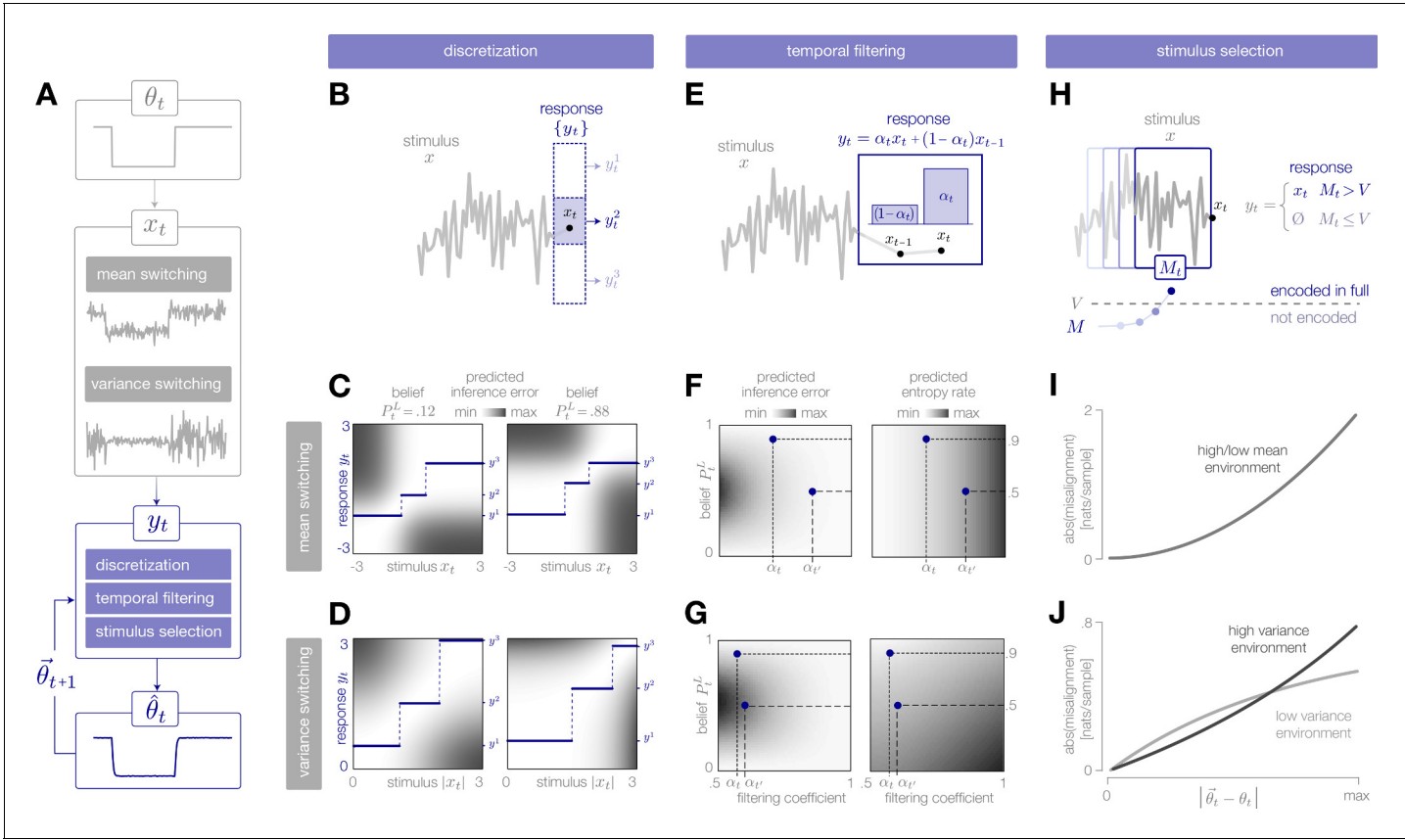

**Figure 2.** Adaptive encoding schemes. (A) We consider a specific implementation of our general framework in which an environmental state $\theta_t$ switches between two values with fixed probability. This state parameterizes the mean or variance of a Gaussian stimulus distribution. Stimuli $x_t$ are drawn from this distribution and encoded in neural responses $y_t$. We consider three encoding schemes that perform discretization (panels B-D), temporal filtering (panels E-G), or stimulus selection (panels H-J) on incoming stimuli. (B) (Schematic) At each timestep, an incoming stimulus $x_t$ (black dot) is mapped onto a discrete neural response level $y_t^i$ (solid blue rectangle) chosen from a set $\{y_t^i\}$ (dotted rectangles). (C–D) The predicted inference error induced by mapping a stimulus $x_t$ onto a neural response $y_t$ varies as a function of the observer's belief $P_t^L$ about the state of the environment (shown for $P_t^L = 0.12$, left column; $P_t^L = 0.88$, right column). At each timestep, the optimal response levels $\{y^1, y^2, y^3\}$ (solid lines) are chosen to minimize this error when averaged over the predicted stimulus distribution. See *Figure 2—figure supplement 1A* for numerical values of color scale. (E) (Schematic) At each timestep, incoming stimuli are combined via a linear filter with a coefficient $\alpha_t$. (F–G) The average predicted inference error (left column) depends on the filter coefficient $\alpha_t$ and on the observer's belief $P_t^L$ about the state of the environment. At each timestep, the optimal filter coefficient (blue dot) is found by balancing error and entropy given a prediction of the environmental state ($\alpha_t$ and $\alpha_{t'}$ are shown for $P_t^L = 0.9$ and $P_{t'}^L = 0.5$, respectively). See *Figure 2—figure supplement 1B* for numerical values of color scale. (H) (Schematic) At each timestep, the encoder computes the misalignment $M_t$ between the predicted and measured surprise of incoming stimuli. If the misalignment exceeds a threshold $V$, the stimulus is encoded with perfect fidelity; otherwise, the stimulus is not encoded. (I–J) The misalignment signal (computed here analytically; see Materials and methods) depends on the relationship between the predicted and true state of the environment. When the mean is changing over time (panel I), the misalignment depends only on the absolute difference between the true and predicted mean. When the variance is changing over time (panel J), the misalignment also depends on the true variance of the environment.

DOI: https://doi.org/10.7554/eLife.32055.005

The following figure supplement is available for figure 2:

**Figure supplement 1.** Minimum and maximum values of the color ranges shown in *Figure 2*.
DOI: https://doi.org/10.7554/eLife.32055.006

cost. Because the encoding is optimized based on the internal prediction of the environmental state, the entropy of the neural response will depend on how closely this prediction aligns with the true state of the environment. The entropy specifies the minimal number of bits required to accurately represent the neural response (*Cover and Thomas, 2012*), and becomes a lower bound on energy expenditure if each bit requires a fixed metabolic cost (*Sterling and Laughlin, 2015*). We therefore use the entropy of the response as a general measure of the metabolic cost of encoding.

We expect efficient encoding schemes to operate on uncertainty and surprise. The observer's uncertainty, given by $H\left[P_t^L\right] = P_t^L\theta^L + \left(1 - P_t^L\right)\theta^H$, is largest when the posterior is near 0.5, and the observer believes that the environment is equally likely to be in either state. The degree to which incoming stimuli are surprising depends on the entropy of the stimulus distribution, and on the alignment between this distribution and the observer's belief. When the mean of the Gaussian distribution is changing in time, the entropy is constant, and surprise depends symmetrically on the squared difference between the true and predicted mean, $(\mu - \vec{\mu})^2$. When the variance is changing, the entropy is also changing in time, and centered surprise depends asymmetrically on the ratio of true and predicted variances, $\sigma^2/\vec{\sigma}^2$. As a result, encoding strategies that rely on stimulus surprise should be symmetric to changes in mean but asymmetric to changes in variance.

To illustrate the dynamic relationship between encoding and inference, we use a 'probe' environment that switches between two states at fixed intervals of $1/h$ timesteps. This specific instantiation is not unlikely given the observer's model of the environment (*DeWeese and Zador, 1998*) and allows us to illustrate average behaviors over many cycles of the environment.

## Encoding via discretization

Neurons use precise sequences of spikes (*Roddey et al., 2000*) or discrete firing rate levels (*Laughlin, 1981*) to represent continuous stimuli. This inherent discreteness imposes a fundamental limitation on the number of distinct neural responses that can be used to represent a continuous stimulus space. Many studies have argued that sensory neurons make efficient use of limited response levels by appropriately tuning these levels to match the steady-state distribution of incoming stimuli (e.g. *Laughlin, 1981*; *Balasubramanian and Berry, 2002*; *Gjorgjieva et al., 2017*).

Here, we consider an encoder that adaptively maps an incoming stimulus $x_t$ onto a discrete set of neural response levels $\left\{y_t^i\right\}$ (*Figure 2B*). Because there are many more stimuli than levels, each level must be used to represent multiple stimuli. The number of levels reflects a global constraint on representational capacity; fewer levels indicates a stronger constraint and results in a lower fidelity encoding.

The encoder can adapt this mapping by expanding, contracting, and shifting the response levels to devote higher fidelity to different regions of the stimulus space. We consider an optimal strategy in which the response levels are chosen at each timestep to minimize the predicted inference error, subject to a constraint on the number of levels:

$$\underbrace{\left\langle \left(\hat{\theta}_{x,t} - \hat{\theta}_{y,t}\right)^2 \right\rangle_{p(x_t|\vec{\theta}_{y,t})}}_{\substack{\text{predicted} \\ \text{inference error}}} \tag{1}$$

When the mean of the stimulus distribution is changing over time, we define these levels with respect to the raw stimulus value $x_t$. When the variance is changing, we define these levels with respect to the absolute deviation from the mean, $|x_t - \mu|$ (where we take $\mu = 0$). The predicted inference error induced by encoding a stimulus $x_t$ in a response $y_t$ changes over time as a function of the observer's prediction of the environmental state (*Figure 2C–D*). Because some stimuli have very little effect on the estimate at a given time, they can be mapped onto the same neural response level without inducing error in the estimate (white regions in *Figure 2C–D*). The optimal response levels are chosen to minimize this error when averaged over the predicted distribution of stimuli.

The relative width of each level is a measure of the resolution devoted to different regions of the stimulus space; narrower levels devote higher resolution (and thus higher fidelity) to the corresponding regions of the stimulus space. The output of these response levels is determined by their alignment with the true stimulus distribution. An encoding that devotes higher resolution to stimuli that are likely to occur in the environment will produce a higher entropy rate (and thus higher cost), because many different response levels will be used with relatively high frequency. In contrast, if an encoding scheme devotes high resolution to surprising stimuli, very few response levels will be used, and the resulting entropy rates will be low.

When designed for accurate inference, we find that the optimal encoder devotes its resolution to stimuli that are surprising given the current prediction of the environment (*Figure 3B*). In a mean-

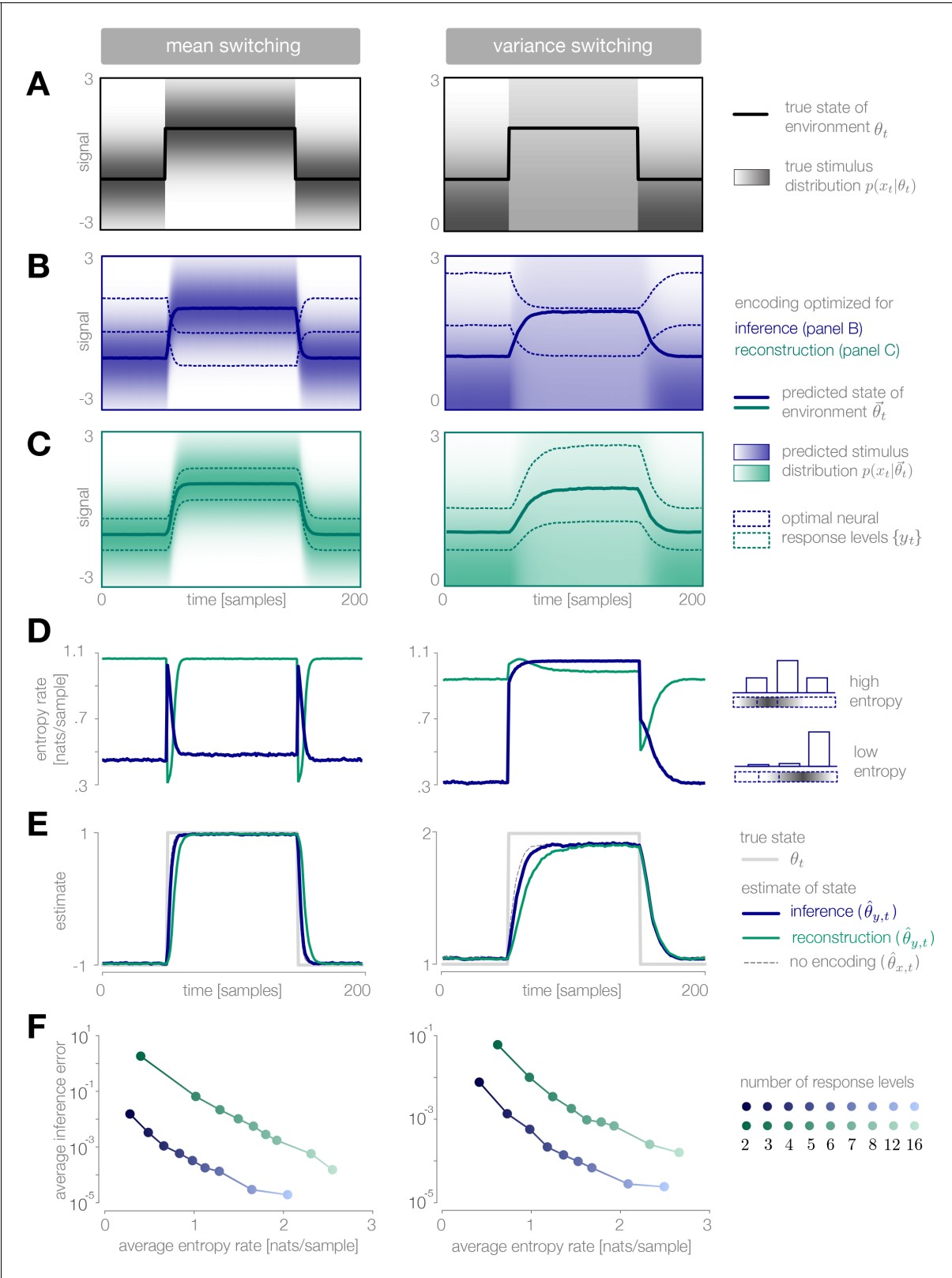

**Figure 3.** Dynamic inference with optimally-adapted response levels. (**A**) We consider a probe environment in which a state $\theta_t$ (solid line) switches between two values at fixed time intervals. This state parametrizes the mean (left) or the variance (right) of a Gaussian stimulus distribution (heatmap). (**B, C**) Optimal response levels (dotted lines) are chosen to minimize error in inference (blue) or stimulus reconstruction (green) based on the predicted

*Figure 3 continued on next page*

*Figure 3 continued*

stimulus distribution $p\left(x_t|\vec{\theta_t}\right)$ (heatmap). Results are shown for three response levels. All probability distributions in panels A-C are scaled to the same range, $[0, 0.4]$. (**B**) Response levels optimized for inference devote higher resolution (narrower levels) to stimuli that are surprising given the current prediction of the environment. (**C**) Response levels optimized for stimulus reconstruction devote higher resolution to stimuli that are likely. (**D**) The entropy rate of the encoding is found by partitioning the true stimulus distribution (heatmap in panel A) based on the optimal response levels (dotted lines in panels B-C). Abrupt changes in the environment induce large changes in entropy rate that are symmetric for mean estimation (left) but asymmetric for variance estimation (right). Apparent differences in the baseline entropy rate for low- versus high-mean states arise from numerical instabilities. (**E**) Encoding induces error in the estimate $\hat{\theta_t}$. Errors are larger if the encoding is optimized for stimulus reconstruction than for inference. The error induced by upward and downward switches is symmetric for mean estimation (left) but asymmetric for variance estimation (right). In the latter case, errors are larger when inferring upward switches in variance. (**F**) Increasing the number of response levels decreases the average inference error but increases the cost of encoding. Across all numbers of response levels, an encoding optimized for inference (blue) achieves lower error at lower cost than an encoding optimized for stimulus reconstruction (green). All results in panels A-C and E are averaged over 500 cycles of the probe environment. Results in panel D were computed using the average response levels shown in panels B-C. Results in panel F were determined by computing time-averages of the results in panels D-E.

DOI: https://doi.org/10.7554/eLife.32055.007

The following figure supplements are available for figure 3:

**Figure supplement 1.** Learning of optimal response levels with Lloyd's algorithm.

DOI: https://doi.org/10.7554/eLife.32055.008

**Figure supplement 2.** Deviations from optimal inference due to transmission noise.

DOI: https://doi.org/10.7554/eLife.32055.009

switching environment (left column of *Figure 3*), stimuli that have high surprise fall within the tails of the predicted stimulus distribution. As a result, when the observer's prediction is accurate, the bulk of the stimulus distribution is mapped onto the same response level (*Figure 3B*, left), and entropy rates are low (blue curve in *Figure 3D*, left). When the environment changes abruptly, the bulk of the new stimulus distribution is mapped onto different response levels. This results in a large spike in entropy rate, which enables the observer to quickly adapt its estimate to the change (blue curve in *Figure 3E*, left).

In a variance-switching environment (right column of *Figure 3*), stimuli that have high centered surprise fall either within the tails of the predicted stimulus distribution (when variance is low), or within the bulk (when variance is high). As a result, entropy rates are low in the low-variance state, but remain high during the high-variance state (blue curve in *Figure 3D*, right).

When designed for accurate reconstruction of the stimulus, we find that the optimal encoder devotes its resolution to stimuli that are likely given the current prediction of the environmental state (*Figure 3C*). As a result, entropy rates are high when the observer's prediction is accurate, regardless of the environment (green curves in *Figure 3D*). Entropy rates drop when the environment changes, because likely stimuli become mapped onto the same response level. This drop slows the observer's detection of changes in the environment (green curve in *Figure 3E*, left). An exception occurs when the variance abruptly increases, because likely stimuli are still given high resolution by the encoder following the change in the environment.

Whether optimizing for inference or stimulus reconstruction, the entropy rate, and thus the coding cost, changes dynamically over time in a manner that is tightly coupled with the inference error. The average inference error can be reduced by increasing the number of response levels, but this induces a higher average coding cost (*Figure 3F*). As expected, a strategy optimized for inference achieves lower inference error than a strategy optimized for stimulus reconstruction (across all numbers of response levels), but it also does so at significantly lower coding cost.

## Encoding via temporal filtering

Neural responses have limited gain and temporal acuity, a feature that is often captured by linear filters. For example, neural receptive fields are often characterized as linear temporal filters, sometimes followed by a nonlinearity (*Bialek et al., 1990*; *Roddey et al., 2000*). The properties of these filters are known to dynamically adapt to changing stimulus statistics (e.g. *Sharpee et al., 2006*; *Sharpee et al., 2011*), and numerous theoretical studies have suggested that such filters are adapted to maximize the amount of information that is encoded about the stimulus (*van Hateren, 1992*; *Srinivasan et al., 1982*).

Here, we consider an encoder that implements a very simple temporal filter (*Figure 2E*):

$$y_t = \alpha_t x_t + (1 - \alpha_t) x_{t-1} \tag{2}$$

where $\alpha_t \in [0.5, 1]$ is a coefficient that specifies the shape of the filter and controls the instantaneous fidelity of the encoding. When $\alpha_t = 0.5$, the encoder computes the average of current and previous stimuli by combining them with equal weighting, and the fidelity is minimal. When $\alpha_t = 1$, the encoder transmits the current stimulus with perfect fidelity (i.e. $y_t = x_t$). In addition to introducing temporal correlations, the filtering coefficient changes the gain of the response $y_t$ by rescaling the inputs $\{x_t, x_{t-1}\}$.

The encoder can adapt $\alpha_t$ in order to manipulate the instantaneous fidelity of the encoding (*Figure 2E*). We again consider an optimal strategy in which the value of $\alpha_t$ is chosen at each time-step to minimize the predicted inference error, subject to a constraint on the predicted entropy rate of the encoding:

$$\underbrace{\left\langle \left( \hat{\theta}_{x,t} - \hat{\theta}_{y,t} \right)^2 \right\rangle_{p(x_t | \vec{\theta}_{y,t})}}_{\substack{\text{predicted} \\ \text{inference error}}} + \underbrace{\beta H(y_t, y_{t+1})}_{\substack{\text{predicted} \\ \text{entropy rate}}} \tag{3}$$

Both terms depend on the strength of averaging $\alpha_t$ and on the observer's belief $P_t^L$ about the state of the environment (*Figure 2F–G*). The inference error depends on belief through the observer's uncertainty; when the observer is uncertain, strong averaging yields a low fidelity representation. When the observer is certain, however, incoming stimuli can be strongly averaged without impacting the observer's estimate. The entropy rate depends on belief through the predicted entropy rate (variance) of the stimulus distribution; when the predicted entropy rate is high, incoming stimuli are more surprising on average. The multiplier $\beta$ reflects a global constraint on representational capacity; larger values of $\beta$ correspond to stronger constraints and reduce the maximum fidelity of the encoding. This, in turn, results in a reduction in coding fidelity through a decrease in gain and an increase in temporal correlation.

When designed for accurate inference, we find that the optimal encoder devotes higher fidelity at times when the observer is uncertain and the predicted stimulus variance is high. In a mean-switching environment, the stimulus variance is fixed (*Figure 4A*, left), and thus the fidelity depends only on the observer's uncertainty. This uncertainty grows rapidly following a change in the environment, which results in a transient increase in coding fidelity (*Figure 4B*, left) and a rapid adaptation of the observer's estimate (*Figure 4D*, left). This estimate is highly robust to the strength of the entropy constraint; even when incoming stimuli are strongly averaged ($\alpha_t = 0.5$), the encoder transmits the mean of two consecutive samples, which is precisely the statistic that the observer is trying to estimate.

In a variance-switching environment, the predicted stimulus variance also changes in time (*Figure 4A*, right). This results in an additional increase in fidelity when the environment is in the high- versus low-variance state, and an asymmetry between the filter responses for downward versus upward switches in variance (*Figure 4B*, right). Both the encoder and the observer are slower to respond to changes in variance than to changes in mean, and the accuracy of the inference is more sensitive to the strength of the entropy constraint (*Figure 4D*, right).

When designed to accurately reconstruct the stimulus, the fidelity of the optimal encoder depends only on the predicted stimulus variance. In a mean-switching environment, the variance is fixed (*Figure 4A*), and thus the fidelity is flat across time. In a variance-switching environment, the fidelity increases with the predicted variance of incoming stimuli, not because variable stimuli are more surprising, but rather because they are larger in magnitude and can lead to higher errors in reconstruction (*Figure 4C*). As the strength of the entropy constraint increases, the encoder devotes proportionally higher fidelity to high-variance stimuli because they have a greater impact on reconstruction error.

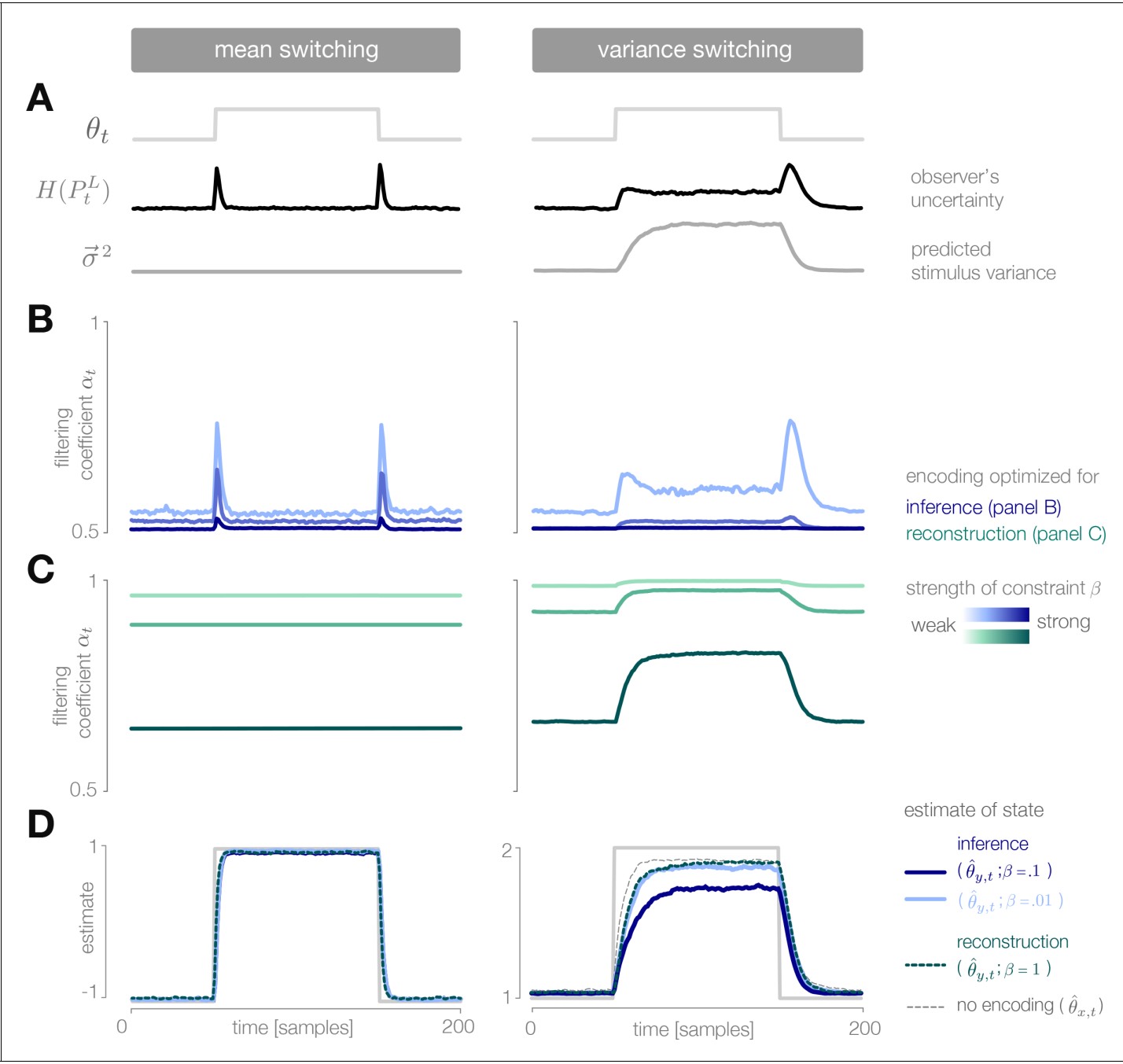

**Figure 4.** Dynamic inference with optimally-adapted temporal filters. (**A**) The observer's uncertainty ($H[P_t^L]$) is largest when the environment is changing. The predicted stimulus variance (a proxy for both the predicted magnitude of the stimulus distribution, and the predicted surprise of incoming stimuli) is constant in a mean-switching environment (left) but variable in a variance-switching environment (right) (computed using a filter coefficient optimized for inference with a weak entropy constraint, corresponding to the lightest blue curves in panel B). (**B, C**) Optimal values of the filter coefficient $\alpha_t$ are chosen at each timestep to minimize error in inference (blue) or stimulus reconstruction (green), subject to a constraint on predicted entropy. Darker colors indicate stronger constraints. (**B**) Filters optimized for inference devote high fidelity at times when the observer is uncertain and stimuli are predicted to be surprising. Shown for $\beta = 0.02, 0.1, 1$ (left) and $\beta = 0.01, 0.1, 1$ (right). (**C**) Filters optimized for reconstruction devote fidelity at times when the magnitude of the stimulus is predicted to be high. Shown for $\beta = 0.01, 0.1, 1$. (**D**) Filtering induces error into the estimate $\hat{\theta}_t$. Strong filtering has minimal impact on mean estimation (left), but induces large errors in the estimation of high variances (right). All results in panels A-D are averaged over 800 cycles of the probe environment.

DOI: https://doi.org/10.7554/eLife.32055.010

## Encoding via stimulus selection

Sensory neurons show sparse activation during natural stimulation (*Vinje and Gallant, 2000*; *Weliky et al., 2003*; *DeWeese and Zador, 2003*), an observation that is often interpreted as a signature of coding cost minimization (*Olshausen and Field, 2004*; *Sterling and Laughlin, 2015*). In particular, early and intermediate sensory neurons may act as gating filters, selectively encoding only highly informative features of the stimulus (*Rathbun et al., 2010*; *Miller et al., 2001*). Such a selection strategy reduces the number of spikes transmitted downstream.

Here, we consider an encoder that selectively transmits only those stimuli that are surprising and are therefore likely to change the observer's belief about the state of the environment. When the observer's prediction is inaccurate, the predicted average surprise $H\left[p\left(x_t|\vec{\theta}_t\right)\right]$ will differ from the true average surprise $H[p(x_t|\theta_t)]$ by an amount equal to the KL-divergence of the predicted from the true stimulus distributions (Materials and methods). In principle, this difference could be used to selectively encode stimuli at times when the observer's estimate is inaccurate.

In practice, however, the encoder does not have access to the entropy of the true stimulus distribution. Instead, it must measure surprise directly from incoming stimulus samples. The measured surprise of each incoming stimulus sample is given by its negative log probability, $-\log\left[p\left(x_t|\vec{\theta}_t\right)\right]$. We consider an encoder that compares the predicted surprise to a running average of the measured surprise. In this way, the encoder can heuristically assess whether a change in the stimulus distribution had occurred by computing the 'misalignment' $M_t$ between the predicted and measured stimulus distributions:

$$M_t = H\left[p\left(x_t|\vec{\theta}_t\right)\right] + \frac{1}{T}\sum_{\tau=0}^{T}\log\left[p\left(x_{t-\tau}|\vec{\theta}_t\right)\right] \tag{4}$$

The misalignment is computed over a time window $T$, which ensures that the observer's prediction does not gradually drift from the true value in cases where surprising stimuli are not indicative of a change in the underlying stimulus distribution (we use $T = 10$). Because the misalignment signal is directly related to the surprise of incoming stimuli, it is symmetric to upward and downward switches in the mean of the stimulus distribution, but it is asymmetric to switches in variance and has a larger magnitude in the high-variance state (shown analytically in *Figure 2I–J*).

The misalignment signal is both non-stationary and non-Gaussian. Optimizing an encoding scheme based on this signal would require deriving the corresponding optimal observer model, which is difficult to compute in the general case. We instead propose a heuristic (albeit sub-optimal) solution, in which the encoder selectively encodes the current stimulus with perfect fidelity ($y_t = x_t$) when recent stimuli are sufficiently surprising and the magnitude of the misalignment signal exceeds a threshold $V$ (*Figure 2H*). When the magnitude of the misalignment signal falls below the threshold, stimuli are not encoded ($y_t = \emptyset$). At these times, the observer does not receive any information about incoming stimuli, and instead marginalizes over its internal prediction to update its estimate (Materials and methods). The value of the threshold reflects a constraint on overall activity; higher thresholds result in stronger criteria for stimulus selection, which decreases the maximum fidelity of the encoding.

When the mean of the stimulus distribution changes in time, very few stimuli are required to maintain an accurate estimate of the environmental state (*Figure 5A–B*, left). When the environment changes abruptly, the observer's prediction is no longer aligned with the environment, and the misalignment signal increases until incoming stimuli are encoded and used to adapt the observer's prediction. Because it requires several stimulus samples for the misalignment to exceed threshold, there is a delay between the switch in the environment and the burst of encoded stimuli. This delay, which is proportional to the size of the threshold, slows the observer's detection of the change (*Figure 5C*, left).

When the variance changes in time, the average surprise of incoming stimuli also changes in time. When the variance abruptly increases, the misalignment signal grows both because the observer's prediction is no longer accurate, and because the average surprise of the incoming stimulus distribution increases. A large proportion of stimuli are transmitted, and the observer quickly adapts to the change. If the threshold is sufficiently high, however, the observer's prediction never fully aligns with the true state. When the variance abruptly decreases, the incoming stimulus distribution is less

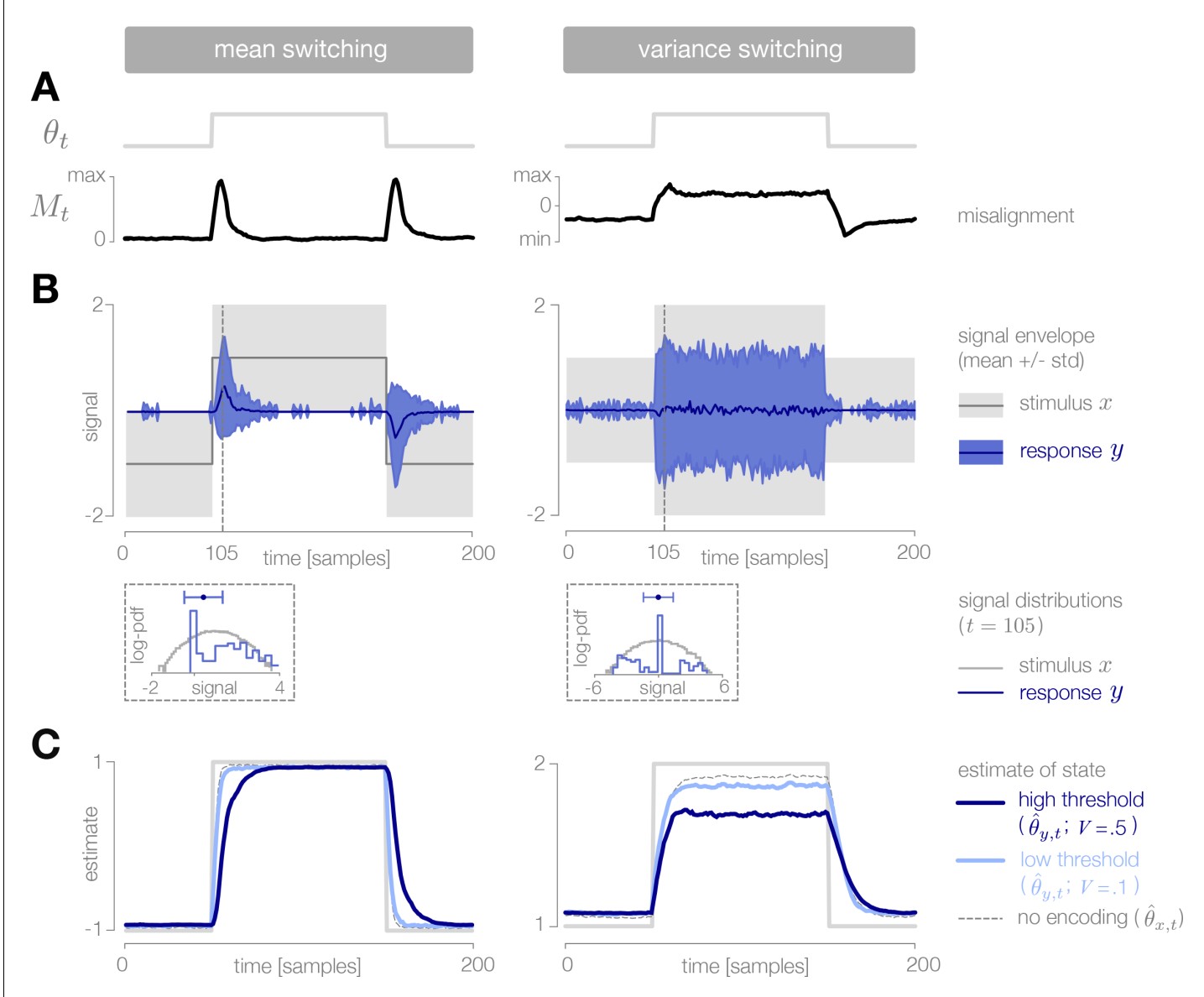

**Figure 5.** Dynamic inference with stimulus selection. (**A**) When the environment is changing, the observer's prediction is misaligned with the state of the environment. When this misalignment $M_t$ is large, stimuli are transmitted in full ($y_t = x_t$). When this misalignment falls below a threshold $V$, stimuli are not transmitted at all ($y_t = 0$). (**B**) The distribution of encoded stimuli changes over time, as can be seen by comparing the envelope of the stimulus distribution (gray) with the envelope of the neural responses (blue). Left: When the mean of the stimulus distribution changes abruptly, a large proportion of stimuli are encoded, and the mean of the neural response (blue line) approaches the mean of the stimulus distribution (black line). At times when the mean of the stimulus distribution is stable, very few stimuli are encoded, and the mean of the neural response drops to zero. Right: When the variance is low, very few stimuli are encoded. When the variance increases, the average surprise of incoming stimuli increases, and a large proportion of stimuli are encoded. The envelope of the neural response expands and approaches the envelope of the stimulus distribution. Insets: At times when the environment is changing (shown for $t = 105$), the distribution of responses (blue) is sparser than the distribution of stimuli (gray), due to the large proportion of stimuli that are not encoded (indicated by the large peak in probability mass at 0). Shown for $V = 0.5$. (**C**) Higher thresholds slow the observer's detection of changes in the mean (left), and cause the observer to underestimate high variances (right). Threshold values are scaled relative to the maximum analytical value of the misalignment signal in the mean- and variance-switching environment (shown in *Figure 2I and J*, respectively). Results in panels B and C are averaged over 800 cycles of the probe environment.

DOI: https://doi.org/10.7554/eLife.32055.011

surprising on average, and therefore a greater number of stimulus samples is needed before the misalignment signal exceeds threshold. As a result, the observer is slower to detect decreases in variance than increases (*Figure 5C*, right).

## Dynamical signatures of adaptive coding

The preceding sections examined the dynamics of optimal encoding strategies as seen through the internal parameters of the encoder itself. The alignment between these internal parameters and the external dynamics of the environment determine the output response properties of each encoder. It is these output response properties that would give experimental access to the underlying encoding scheme, and that could potentially be used to distinguish an encoding scheme optimized for inference from one optimized for stimulus reconstruction.

To illustrate this, we simulate output responses of each encoder to repeated presentations of the probe environment. In the case of discretization, we use a simple entropy coding procedure to map each of four response levels to four spike patterns ($[00], [01], [10], [11]$) based on the probability that each response level will be used given the distribution of incoming stimuli, and we report properties of the estimated spike rate (see spike rasters in *Figure 6A*; Materials and methods). In the cases of filtering and stimulus selection, we report properties of the response $y_t$.

We find that encodings optimized for inference typically show transient changes in neural response properties after a switch in the environment, followed by a return to baseline. This is manifested in a burst in firing rates in the case of discretization, and a burst in response variability in the cases of filtering and stimulus selection. Filtering is additionally marked by a transient decrease in the temporal correlation of the response. The magnitude of these transient changes relative to baseline is most apparent in the case of mean estimation, where the variability in the environment remains fixed over time. Because periods of higher variability in the environment are intrinsically more surprising, baseline response properties change during variance estimation, and bursts relative to baseline are less pronounced. Nevertheless, we see a transient decrease in temporal correlation in the case of filtering, and a transient increase in response variability in the case of stimulus selection, following switches in variance.

The same dynamical features are not observed in encoders optimized for stimulus reconstruction. For mean estimation, firing rates and response variability remain nearly constant over time, despite abrupt changes in the mean of the stimulus distribution. Discretization shows a brief rise and dip in firing rate following a switch, which has been observed experimentally (*Fairhall et al., 2001*). For variance estimation, response properties show sustained (rather than transient) changes following a switch.

Differences in response properties are tightly coupled to the speed and accuracy of inference, as mediated by the feedforward and feedback interactions between the encoder and the observer. Note that these measures of speed and accuracy (as well as the comparisons made in *Figures 3E*, *4D,* and *5C*) intrinsically favor encodings optimized for inference; we therefore restrict our comparison to this set of encodings. We find that both the speed and accuracy of inference are symmetric to changes in the mean of the stimulus distribution, but asymmetric to changes in variance. This is qualitatively consistent with the optimal Bayesian observer in the absence of encoding (*DeWeese and Zador, 1998*). We find that encoding schemes optimized for inference have a more significant impact on the speed and accuracy of variance estimation than of mean estimation. Interestingly, the speed of variance adaptation deviates from optimality in a manner that could potentially be used to distinguish between encoding strategies. In the absence of encoding, the ideal observer is faster to respond to increases than to decreases in variance. We find that encoding via stimulus selection increases this asymmetry, encoding via discretization nearly removes this asymmetry, and encoding via stimulus selection reverses this asymmetry.

Together, these observations suggest that both the dynamics of the neural response and the patterns of deviation from optimal inference could be used to infer features of the underlying sensory coding scheme. Moreover, these results suggest that an efficient system could prioritize some encoding schemes over others, depending on whether the goal is to reconstruct the stimulus or infer its underlying properties, and if the latter, whether this goal hinges on speed, accuracy, or both.

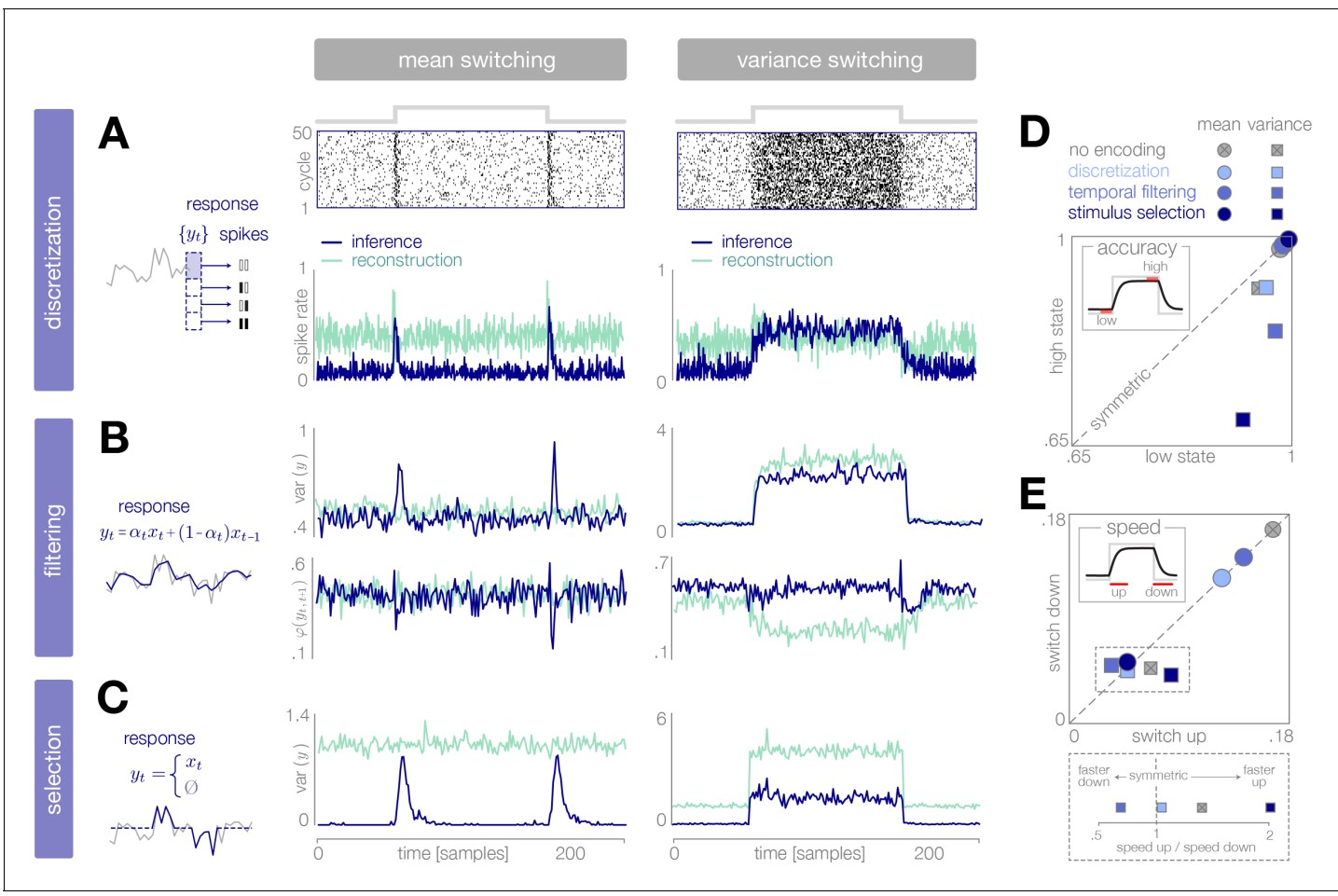

**Figure 6.** Dynamical signatures of adaptive coding schemes. (A–C) We simulate the output of each encoder to repeated cycles of the probe environment. In the case of discretization (panel A), we use a simple entropy coding procedure to map optimal response levels $\{y_t\}$ onto spike patterns, as shown by the spike rasters. In the case of temporal filtering (panel B) and stimulus selection (panel C), we measure properties of the response $y_t$. When encodings are optimized for inference (dark blue traces), abrupt changes in the mean of the stimulus distribution (panels A-C, left) are followed by transient increases in spike rate (discretization, panel A) and response variability (filtering, panel B; stimulus selection, panel C). In the case of temporal filtering, these changes are additionally marked by decreases in the temporal correlation $\varphi(y_{t,t+1})$ of the response. In contrast, the response properties of encoders optimized for stimulus reconstruction (light green traces) remain more constant over time. Abrupt changes in variance (panels A-C, right) are marked by changes in baseline response properties. Responses show transient deviations away from baseline when encodings are optimized for inference, but remain fixed at baseline when encodings are optimized for reconstruction. In all cases, encodings optimized for inference maintain lower baseline firing rates, lower baseline variability, and higher baseline correlation than encodings optimized for stimulus reconstruction. Spike rates (panel A) are averaged over 500 cycles of the probe environment. Response variability (panels B-C) is computed at each timepoint across 800 cycles of the probe environment. Temporal correlation $\varphi(y_{t,t+1})$ (panel B) is computed between consecutive timepoints across 800 cycles of the probe environment. (D–E) Encoding schemes impact both the accuracy (panel D) and speed (panel E) of inference. In all cases, the dynamics of inference are symmetric for changes in mean (points lie along the diagonal) but asymmetric for changes in variance (points lie off the diagonal). Encodings decrease the accuracy of estimating high-variance states (panel D), and they alter the speed of responding to changes in both mean and variance. The response to upward versus downward switches (dotted box) separates encoding schemes based on whether they are faster (right of dotted vertical line) or slower (left of dotted vertical line) to respond to increases versus decreases in variance. Speed and accuracy are measured from the trial-averaged trajectories of $\hat{\theta}_{y,t}$ shown in *Figure 3E, Figure 4D* ($\beta = 0.01$), and *Figure 5C* ($V = 0.5$) (Materials and methods).

DOI: https://doi.org/10.7554/eLife.32055.012

The following figure supplement is available for figure 6:

**Figure supplement 1.** Physically different stimuli become indistinguishable to an adapted system optimized for inference.

DOI: https://doi.org/10.7554/eLife.32055.013

## Adaptive coding for inference under natural conditions

The simplified task used in previous sections allowed us to explore the dynamic interplay between encoding and inference. To illustrate how this behavior might generalize to more naturalistic settings, we consider a visual inference task with natural stimuli (*Figure 7A*, Materials and methods). In particular, we model the estimation of variance in local curvature in natural image patches—a computation similar to the putative function of neurons in V2 (*Ito and Komatsu, 2004*). As before, the goal of the system is to infer a change in the statistics of the environment from incoming sensory stimuli. We consider a sequence of image patch stimuli drawn randomly from a local region of a natural image; this sequence could be determined by, for example, saccadic fixations. Each image patch is encoded in the responses of a population of sensory neurons using a well-known sparse-coding model (*Olshausen and Field, 1996*). After adapting to natural stimulus statistics, the basis functions of each model neuron resemble receptive fields of simple cells in V1. A downstream observer decodes the stimulus from this population response and normalizes its contrast. The contrast-normalized patch is then projected onto a set of curvature filters. The variance in the output of these filters is used as an estimate of the underlying statistics of the image region. Both the computation of local image statistics and visual sensitivity to curvature are known to occur in V2 (*Freeman et al., 2013*; *Ito and Komatsu, 2004*; *Yu et al., 2015*).

The encoder reconstructs each stimulus subject to a sparsity constraint $\lambda$; large values of $\lambda$ decrease the population activity at the cost of reconstruction accuracy (*Figure 7—figure supplement 1*). In contrast to the encoding models discussed previously, this encoder is explicitly optimized to reconstruct each stimulus, rather than to support accurate inference. Even in this scenario, however, the observer can manipulate the sparsity of the population response to decrease resource use while maintaining an accurate estimate of the environmental state. It has been proposed that early sensory areas, such as V1, could manipulate the use of metabolic resources depending on top-down task demands (e.g. *Rao and Ballard, 1999*).

We model a change in the stimulus distribution by a gaze shift from one region of the image to another (*Figure 7B*). This shift induces an increase in the variance of curvature filters. Following this change, the observer must update its estimate of local curvature using image patches drawn from the new image region. We empirically estimated the impact of stimulus surprise and observer uncertainty on this estimation and found it to be consistent with results based on model environments (*Figure 7D*; compare with *Figure 1B*). Surprising stimuli that project strongly on curvature filters exert a large impact on inference, while expected stimuli (characterized by low centered surprise) exert little impact (*Figure 7C–D, F*). Similarly, individual stimuli exert a larger impact on the estimate when the observer is uncertain than when the observer is certain (*Figure 7D–E*).

The system can modulate the sparsity of the population response based on uncertainty and surprise. To illustrate this, we simulated neural population activity in response to a change in each of these quantities (*Figure 7E and F*, respectively). To do this, we selected a sequence of 45 image patches, 5 of which were chosen to have high centered surprise (*Figure 7F*; red marker) or to correspond to an observer with high uncertainty (*Figure 7E*; red marker). An increase in either surprise or uncertainty requires a higher fidelity response to maintain an approximately constant level of inference error. This results in a burst of population activity (blue traces in *Figure 7E–F*). Similar population bursts were recently observed in V1 in response to violations of statistical regularities in stimulus sequences (*Homann et al., 2017*). When optimized for constant reconstruction error, the sparsity of the population response remains fixed in time. The resulting population response does not adapt, and instead fluctuates around a constant value determined by $\lambda$ (green traces in *Figure 7E–F*).

## Discussion

Organisms rely on incoming sensory stimuli to infer behaviorally relevant properties of their environment, and hierarchical inference is postulated to be a computational function of a broad range of neural circuits (*Lee and Mumford, 2003*; *Fiser et al., 2010*). Representing and transmitting these stimuli, however, is energetically costly, and such costs are known to constrain the design and function of the nervous system (*Sterling and Laughlin, 2015*). Here, we explored the interplay between efficient encoding and accurate inference, and we identified two general principles that can be used to balance these objectives. First, when the environment is changing over time, the relative utility of

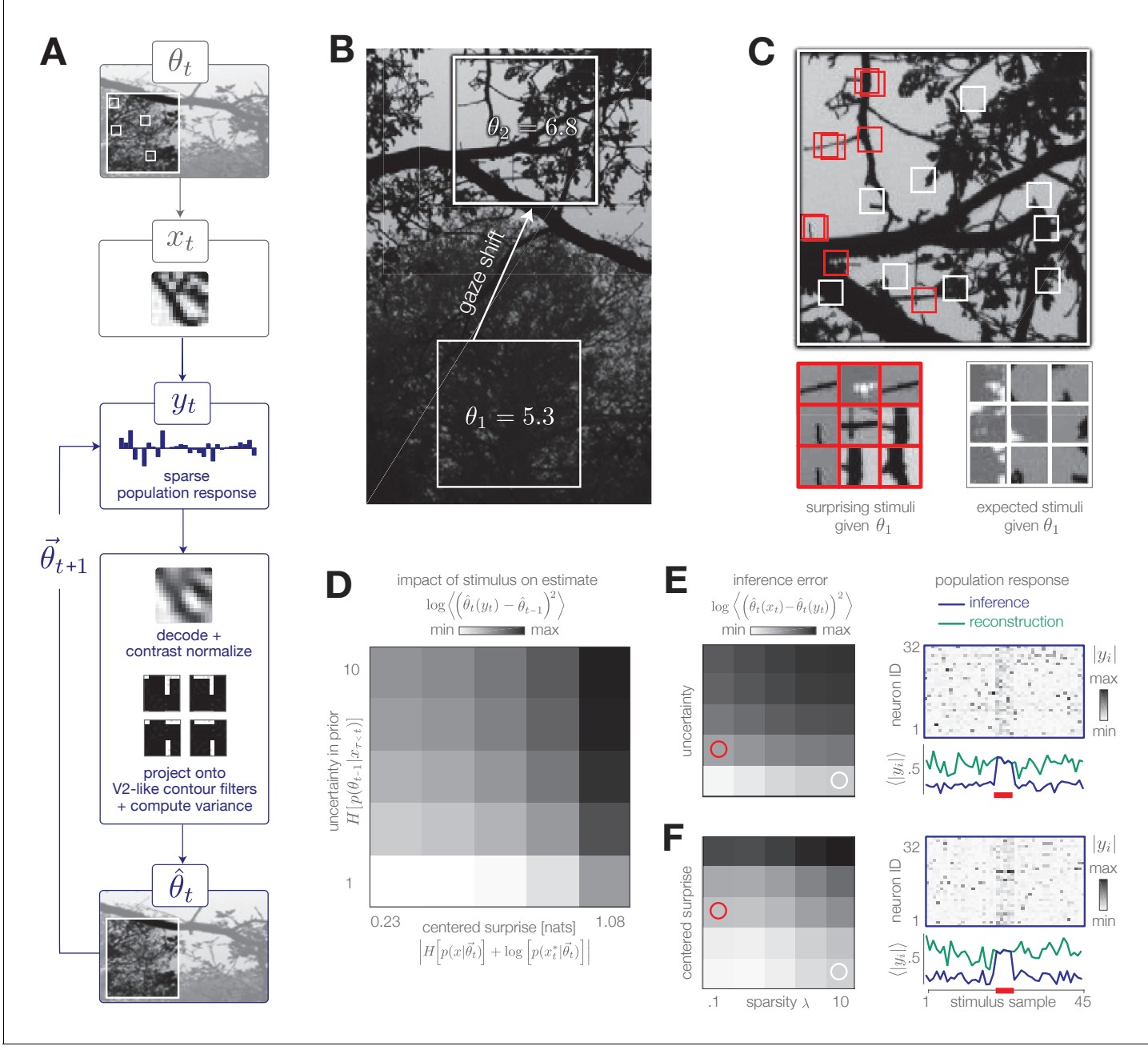

**Figure 7.** Model inference task with natural stimuli. (**A**) We model a simple task of inferring the variance of local curvature in a region of an image. The system encodes randomly drawn image patches that model saccadic fixations. Individual image patches are encoded in sparse population activity via V1-like receptive fields (see *Figure 7—figure supplement 1*). Image patches are then decoded from the population activity, contrast-normalized, and projected onto V2-like curvature filters. The observer computes the variance of these filter outputs. (**B**) After a gaze shift from an area of low curvature (bottom square, $\theta = \theta_1$) to an area of high curvature (top square, $\theta = \theta_2$), the observer must update its estimate of local curvature. (**C**) Image patches that are surprising given the observer's estimate (red) have larger variance in curvature, while expected patches (white) have low variance in curvature. Frames of highly overlapping patches were slightly shifted for display purposes. (**D**) Individual image patches have a large impact on the observer's estimate when the observer is uncertain and when image patches have high centered surprise, analogous to the behavior observed in simple model environments (see *Figure 1B*). Shown for $\lambda = 0.1$. Impact spans the interval [0, 34.12]. (**E**) The observer can exploit its uncertainty to adapt the sparsity of the sensory encoding (heatmap; blue trace). When the observer is certain (white marker), population activity can be significantly reduced without changing the inference error. Increases in uncertainty (red marker) result in bursts of activity (red bar). An encoder optimized for constant reconstruction error produces activity that remains constant over time (green trace). Inference error spans the interval [0, 2.22]. (**F**) The observer can similarly exploit the predicted surprise of incoming stimuli to reduce population activity when stimuli are expected. Inference error spans the interval [0, 1.57].
DOI: https://doi.org/10.7554/eLife.32055.014

*Figure 7 continued on next page*

*Figure 7 continued*

The following figure supplement is available for figure 7:

**Figure supplement 1.** Sparse coding model of natural image patches.
DOI: https://doi.org/10.7554/eLife.32055.015

incoming stimuli for inference can also change. Second, physically different signals can exert similar influence on the observer's model of the environment and can therefore be encoded in the same neural representation without negatively affecting the inference process.

We introduced a general theoretical framework that could exploit these two principles in order to dynamically reduce metabolic costs while maintaining accurate inferences about the environment. This framework employs a well-known computational motif consisting of a feedback loop between an observer and an encoder. We demonstrated that when the goal is accurate inference, the encoder can optimally adapt depending on the uncertainty in the observer's belief about the state of the environment, and on the surprise of incoming stimuli given this belief. This optimal adaptation enables the system to efficiently infer high-level features from low-level inputs, which we argue is a broad goal of neural circuits across the brain. We therefore expect this framework to bear relevance for many different stages of sensory processing, from the periphery through the midbrain to central brain areas.

## Transient increases in fidelity signal salient changes in the environment

To maintain low metabolic costs, we found that encoders optimized for inference adapt their encoding strategies in response to the changing utility of incoming stimuli. This adaptation was signaled by elevated periods of response variability, temporal decorrelation, or total activity. Transient, burst-like changes in each of these properties served to increase the fidelity of the neural response, and enabled the system to quickly respond to informative changes in the stimulus distribution. In the nervous system, bursts of high-frequency activity are thought to convey salient changes in an organism's surroundings (*Marsat et al., 2012*). For example, in the lateral line lobe of the weakly electric fish, neurons burst in response to electric field distortions similar to those elicited by prey (*Oswald et al., 2004*), and these bursts are modulated by predictive feedback from downstream neurons (*Marsat et al., 2012*). Similarly, in the auditory system of the cricket, bursts signal changes in frequency that are indicative of predators, and the amplitude of these bursts is closely linked to the amplitude of behavioral responses (*Sabourin and Pollack, 2009*; *Marsat and Pollack, 2006*). In the visual system, retinal ganglion cells fire synchronously in response to surprising changes in the motion trajectory of a stimulus (*Schwartz et al., 2007*), and layer 2/3 neurons in primary visual cortex show transient elevated activity in response to stimuli that violate statistical regularities in the environment (*Homann et al., 2017*). Neurons in IT cortex show strong transient activity in response to visual stimuli that violate predicted transition rules (*Meyer and Olson, 2011*), and recent evidence suggests that single neurons in IT encode latent probabilities of stimulus likelihood during behavioral tasks (*Bell et al., 2016*). In thalamus, burst firing is modulated by feedback from cortex (*Halassa et al., 2011*) and is thought to signal the presence of informative stimuli (*Lesica and Stanley, 2004*; *Miller et al., 2001*; *Rathbun et al., 2010*). In the auditory forebrain of the zebra finch, neural activity is better predicted by the surprise of a stimulus than by its spectrotemporal content (*Gill et al., 2008*), and brief synchronous activity is thought to encode a form of statistical deviance of auditory stimuli (*Beckers and Gahr, 2012*). We propose that this broad range of phenomena could be indicative of an active data selection process controlled by a top-down prediction of an incoming stimulus distribution, and could thus serve as an efficient strategy for encoding changes in the underlying statistics of the environment. While some of these phenomena appear tuned to specific stimulus modulations (such as those elicited by specific types of predators or prey), we argue that transient periods of elevated activity and variability more generally reflect an optimal strategy for efficiently inferring changes in high-level features from low-level input signals.

In some cases, it might be more important to reconstruct details of the stimulus itself, rather than to infer its underlying cause. In such cases, we found that the optimal encoder maintained consistently higher firing rates and more heterogeneous response patterns. In both the cricket (*Sabourin and Pollack, 2010*) and the weakly electric fish (*Marsat et al., 2012*), heterogeneous

neural responses were shown to encode stimulus details relevant for evaluating the quality of court-ship signals (in contrast to the bursts of activity that signal the presence of aggressors). While separate circuits have been proposed to implement these two different coding schemes (inferring the presence of an aggressor versus evaluating the quality of a courtship signal), these two strategies could in principle be balanced within the same encoder. The signatures of adaptation that distinguish these strategies could alternatively be used to identify the underlying goal of a neural encoder. For example, neurons in retina can be classified as 'adapting' or 'sensitizing' based on the trajectory of their firing rates following a switch in stimulus variance (*Kastner and Baccus, 2011*). These trajectories closely resemble the response entropies of encoders optimized for inference or reconstruction, respectively (right panel of *Figure 3D*). A rigorous application of the proposed framework to the identification of neural coding goals is a subject of future work.

Importantly, whether the goal is inference or stimulus reconstruction, the encoders considered here were optimized based on predictive feedback from a downstream unit and thus both bear similarity to hierarchical predictive coding as formulated by *Rao and Ballard (1999)*. The goal, however, crucially determines the difference between these strategies: sustained heterogeneous activity enables reconstruction of stimulus details, while transient bursts of activity enable rapid detection of changes in their underlying statistics.

## Periods of stationarity give rise to ambiguous stimulus representations

A central idea of this work is that stimuli that are not useful for a statistical estimation task need not be encoded. This was most notably observed during periods in which an observer maintained an accurate prediction of a stationary stimulus distribution. Here, different stimuli could be encoded by the same neural response without impacting the accuracy of the observer's prediction. This process ultimately renders stimuli ambiguous, and it predicts that the discriminability of individual stimuli should decrease over time as the system's internal model becomes aligned with the environment (Materials and methods, *Figure 6—figure supplement 1*). Ambiguous stimulus representation have been observed in electrosensory pyramidal neurons of the weakly electric fish, where adaptation to the envelope of the animal's own electric field (a second-order statistic analogous to the variance step considered here) reduces the discriminability of specific amplitude modulations (*Zhang and Chacron, 2016*). Similarly, in the olfactory system of the locust, responses of projection neurons to chemically similar odors are highly distinguishable following an abrupt change in the odor environment, but become less distinguishable over time (*Mazor and Laurent, 2005*). The emergence of ambiguous stimulus representations has recently been observed in human perception of auditory textures that are generated from stationary sound sources such as flowing water, humming wind, or large groups of animals (*McDermott et al., 2013*). Human listeners are readily capable of distinguishing short excerpts of sounds generated by such sources. Surprisingly, however, when asked to tell apart long excerpts of auditory textures, performance sharply decreases. We propose that this steady decrease in performance with excerpt duration reflects adaptive encoding for accurate inference, where details of the stimulus are lost over time in favor of their underlying statistical summary.

## Efficient use of metabolic resources yields diverse signatures of suboptimal inference

We used an ideal Bayesian observer to illustrate the dynamic relationship between encoding and inference. Ideal observer models have been widely used to establish fundamental limits of performance on different sensory tasks (*Geisler et al., 2009*; *Geisler, 2011*; *Weiss et al., 2002*). The Bayesian framework in particular has been used to identify signatures of optimal performance on statistical estimation tasks (*Simoncelli, 2009*; *Robert, 2007*), and a growing body of work suggests that neural systems explicitly perform Bayesian computations (*Deneve, 2008*; *Fiser et al., 2010*; *Ma et al., 2006b*; *Rao et al., 2002*). In line with recent studies (*Wei and Stocker, 2015*; *Ganguli and Simoncelli, 2014*), we examined the impact of limited metabolic resources on such probabilistic neural computations.

While numerous studies have identified signatures of near-optimal performance in both neural coding (e.g. *Wark et al., 2009*) and perception (e.g. *Burge and Geisler, 2015*; *Weiss et al., 2002*), the ideal observer framework can also be used to identify deviations from optimality. Such deviations have been ascribed to noise (*Geisler, 2011*) and suboptimal neural decoding (*Putzeys et al.,*

*2012*). Here, we propose that statistical inference can deviate from optimality as a consequence of efficient, resource-constrained stimulus coding. We observed deviations from optimality in both the speed and accuracy of inference, and we found that some of these deviations (namely asymmetries in the speed of variance adaptation) could potentially be used to differentiate the underlying scheme that was used to encode incoming stimuli. It might therefore be possible to infer underlying adaptation strategies by analyzing patterns of suboptimal inference.

## Limitations and future work

We discussed general principles that determine optimal encoding strategies for accurate inference, and we demonstrated the applicability of these principles in simple model systems. Understanding the applicability in more complex settings and for specific neural systems requires further investigation.

### Complexity of the environment

We considered a simple nonstationary environment whose dynamics varied on a single timescale. These dynamics were parameterized by a single latent variable that specified either the mean or the variance of a Gaussian stimulus distribution. These first- and second-order moments are basic properties of an input distribution and often correspond to interpretable, physical properties such as luminance or local contrast. Similar stimulus distribution have been used to study a range of neural and perceptual dynamics, including adaptation of fly visual neurons to changes in luminance and contrast (*Fairhall et al., 2001*), neural representations of electric field modulations in the weakly electric fish (*Zhang and Chacron, 2016*), and human perceptual decision making (*Nassar et al., 2010*). Here, we used this simple environment to probe the dynamics of encoding schemes optimized for inference. We found that optimal encoding schemes respond strongly to changes in the underlying environmental state, and thereby carry information about the timescale of environmental fluctuations. In natural settings, signals vary over a range of temporal scales, and neurons are known to be capable of adapting to multiple timescales in their inputs (*Lundstrom et al., 2008*; *Wark et al., 2009*). We therefore expect that more complex environments, for example those in which the environmental state can both switch between distinct distributions and fluctuate between values within a single distribution, will require that the encoder respond to environmental changes on multiple timescales.

In all such cases, we expect the dimensionality of the latent variable space to determine the lower bound on coding costs for inference. Even in the limit of highly complex models, however, we expect accurate inference and reconstruction to impose qualitatively different constraints on neural response properties.

### Diversity of sensory encoding schemes

We considered three encoding schemes that approximate known features of neural responses, and as such could be implemented broadly across the brain. Discretization is a non-linear encoding scheme that specifies a finite set of instantaneous response levels (such as spiking patterns or discriminable firing rates) and provides a good model of retinal ganglion cells responses (e.g. *Koch et al., 2004*). Temporal filtering, on the other hand, is a linear encoding scheme that forms the basis of a broad class of linear-nonlinear (LN) models. These models have been used to describe neural responses in a range of systems (*Sharpee, 2013*), and can capture temporal dependencies in the neural response. To more closely approximate spiking nonlinearities observed in real neurons, the linear output of this encoder could be followed by a nonlinearity whose parameters are also adapted over time, thereby enabling the system to more strongly suppress irrelevant stimuli. Finally, our model of stimulus selection implements a form of gating, whereby unsurprising stimuli are not encoded. This nonlinear encoding scheme produces bimodal responses (either strongly active or completely silent), and we would therefore expect such a mechanism to be useful when transmitting signals over long distances. This scheme can also be viewed as implementing a partitioning of the stimulus space into surprising and unsurprising stimuli, similar to discretization.

In order to achieve optimal bounds on performance, the parameters of each encoding scheme were computed and updated on each timestep. While it is known that neural systems can adapt on

timescales approaching physical limits (*Fairhall et al., 2001*), it is possible that more complex neural circuits might implement a heuristic version of this adaptation that operates on slower timescales.

Together, these approaches provide a framework for studying adaptive coding across a broad class of neural encoding schemes. This framework can be implemented with other encoding schemes, such as population or spike-time coding. In such cases, we expect that the principles identified here, including increased coding fidelity during periods of uncertainty or surprise, will generalize across encoding schemes to determine optimal strategies of adaptation.

## Robustness to noise

Noise can arise at different stages of neural processing and can alter the faithful encoding and transmission of stimuli to downstream areas (*Roddey et al., 2000*; *Brinkman et al., 2016*). Individual neurons and neural populations can combat the adverse effects of noise by appropriately tuning their coding strategies, for example by adjusting the gain or thresholds of individual neurons (*van Hateren, 1992*; *Gjorgjieva et al., 2017*), introducing redundancies between neural responses (*Doi and Lewicki, 2014*; *Tkacik et al., 2010*; *Moreno-Bote et al., 2014*; *Abbott and Dayan, 1999*; *Sompolinsky et al., 2001*), and forming highly distributed codes (*Denève and Machens, 2016*; *Deneve and Chalk, 2016*). Such optimal coding strategies depend on the source, strength, and structure of noise (*Brinkman et al., 2016*; *Tkacik et al., 2010*; *van Hateren, 1992*; *Kohn et al., 2016*), and can differ significantly from strategies optimized in the absence of noise (*Doi and Lewicki, 2014*).

Noise induced during encoding stages can affect downstream computations, such as the class of inference tasks considered here. To examine its impact on optimal inference, we injected additive Gaussian noise into the neural response transmitted from the discretizing encoder to the observer. We found that the accuracy of inference was robust to low levels of noise, but degraded quickly once the noise variance approached the degree of separation between environmental states (*Figure 3—figure supplement 2*). Although this form of Gaussian transmission noise was detrimental to the inference process, previous work has argued that noise-related variability, if structured appropriately across *a population* of encoders, could support representations of the probability distributions required for optimal inference (*Ma et al., 2006a*). Moreover, we expect that the lossy encoding schemes developed here could be beneficial in combating noise injected *prior* to the encoding step, as they can guarantee that metabolic resources are not wasted in the process of representing noise fluctuations.

Ultimately, the source and degree of noise can impact both the goal of the system and the underlying coding strategies. Here, we considered the goal of optimally inferring changes in environmental states. However, in noisy environments where the separation between latent environmental states is low, a system might need to remain stable in the presence of noise, rather than flexible to environmental changes. We expect that the optimal balance between stability and flexibility to be modulated by the spread of the stimulus distribution relative to the separation between environmental states. A thorough investigation of potential sources of noise, and their impact on the balance between efficient coding and optimal inference, is the subject of future work.

## Measures of optimal performance

To measure the optimal bound on inference error, we used the mean squared difference between point estimates derived in the presence and absence of an encoding step. This metric is general and makes no assumptions about the form of the posterior distribution (*Jaynes, 2003*; *Robert, 2007*). Other measures, such as KL-divergence, could be used to capture not only changes in point estimates, but also changes in uncertainty underlying these estimates.

## Connections to existing theoretical frameworks

Efficient coding of task-relevant information has been studied before, primarily within the framework of the Information Bottleneck (IB) method (*Tishby et al., 2000*; *Chechik et al., 2005*; *Strouse and Schwab, 2016*). The IB framework provides a general theoretical approach for extracting task-relevant information from sensory stimuli, and it has been successfully applied to the study of neural coding in the retina (*Palmer et al., 2015*) and in the auditory cortex (*Rubin et al., 2016*). In parallel, Bayesian Efficient Coding (BEC) has recently been proposed as a framework through which a

metabolically-constrained sensory system could minimize an arbitrary error function that could, as in IB, be chosen to reflect task-relevant information (*Park and Pillow, 2017*). However, neither framework (IB nor BEC) explicitly addresses the issue of adaptive sensory coding in non-stationary environments, where the relevance of different stimuli can change in time. Here, we frame general principles that constrain the dynamic balance between coding cost and task relevance, and we pose neurally plausible implementations.

Our approach bears conceptual similarities to the predictive coding framework proposed by *Rao and Ballard (1999)*, in which low-level sensory neurons support accurate stimulus reconstruction by encoding the residual error between an incoming stimulus and a top-down prediction of the stimulus. Our encoding schemes similarly use top-down predictions to encode useful deviations in the stimulus distribution. Importantly, however, the goal here was not to reconstruct the stimulus itself, but rather to infer the underlying properties of a changing stimulus distribution. To this end, we considered encoding schemes that could use top-down predictions to adaptively adjust their strategies over time based on the predicted utility of different stimuli for supporting inference.

This work synthesizes different theoretical frameworks in an effort to clarify their mutual relationship. In this broad sense, our approach aligns with recent studies that aim to unify frameworks such as efficient coding and Bayesian inference (*Park and Pillow, 2017*), as well as concepts such as efficient, sparse, and predictive coding (*Chalk et al., 2017*).

## Outlook

Efficient coding and probabilistic inference are two prominent frameworks in theoretical neuroscience that address the separate questions of how stimuli can be encoded at minimal cost, and how stimuli can be used to support accurate inferences. In this work, we bridged these two frameworks within a dynamic setting. We examined optimal strategies for encoding sensory stimuli while minimizing the error that such encoding induces in the inference process, and we contrasted these with strategies designed to optimally reconstruct the stimulus itself. These two goals could correspond to different regimes of the same sensory system (*Balasubramanian et al., 2001*), and future work will explore strategies for balancing these regimes depending on task requirements. In order to test the implications of this work for physiology and behavior, it will be important to generalize this framework to more naturalistic stimuli, noisy encodings, and richer inference tasks. At present, our results identify broad signatures of a dynamical balance between metabolic costs and task demands that could potentially explain a wide range of phenomena in both neural and perceptual systems.

# Materials and methods

## A. Optimal Bayesian inference with adaptively encoded stimuli

We describe a class of discrete-time environmental stimuli $x_t$ whose statistics are completely characterized by a single time-varying environmental state variable $\theta_t$.

We then consider the scenario in which these stimuli are encoded in neural responses, and it is these neural responses that must be used to construct the posterior probability over environmental states. In what follows, we derive the optimal Bayesian observer for computing this posterior given the history of neural responses. The steps of this estimation process are summarized in *Figure 1—figure supplement 1*.

In a full Bayesian setting, the observer should construct an estimate of the stimulus distribution, $p(x_t)$, by marginalizing over its uncertainty in the estimate of the environmental state $\theta_t$ (i.e. by computing $p(x_t) = \int d\theta_t\, p(x_t|\theta_t) p(\theta_t)$). For simplicity, we avoid this marginalization by assuming that the observer's belief is well-summarized by the average of the posterior, which is captured by the point value $\hat{\theta}_t = \int d\theta_t\, \theta_t p(\theta_t)$ for estimation, and $\vec{\theta}_{t+1} = \int d\theta_{t+1}\, \theta_{t+1} p(\theta_{t+1})$ for prediction. The average of the posterior is an optimal scalar estimate that minimizes the mean squared error between the estimated and true states of the environment, and is known to provide a good description of both neural (*DeWeese and Zador, 1998*) and perceptual (*Nassar et al., 2010*) dynamics. The observer then uses these point values to condition its prediction of the stimulus distribution, $p\left(x_t|\vec{\theta}_t\right)$. Conditioning on a point estimate guarantees that the observer's prediction of the environment belongs to the same family of distributions as the true environment. This is not guaranteed to be the case when

marginalizing over uncertainty in $\theta_t$. For example, if the posterior assigns non-zero probability mass to two different mean values of a unimodal stimulus distribution, the predicted stimulus distribution could be bimodal, even if the true stimulus distribution is always unimodal. We verified numerically that the key results of this work are not affected by approximating the full marginalization with point estimates.

When the timescale of the environment dynamics is sufficiently slow, the point prediction $\vec{\theta}_{t+1}$ can be approximated by the point estimate $\hat{\theta}_t$. In the two-state environments considered here, the probability that the environment remains in the low state from time $t$ to time $t+1$ is equal to $P_{t+1}^L = P_t^L(1-h) + (1-P_t^L)h$, where $h$ is the hazard rate (*DeWeese and Zador, 1998*). For the small hazard rate used here ($h = 0.01$), $P_{t+1}^L = 0.99P_t^L + 0.01(1-P_t^L)$, and the estimate $\hat{\theta}_t$ is therefore a very close approximation of the prediction $\vec{\theta}_{t+1}$. All results presented in the main text were computed using this approximation (i.e. $\vec{\theta}_{t+1} \approx \hat{\theta}_t$). With this approximation, the optimal Bayesian observer computes the approximate posterior distribution $p\left(\theta_t | y_{\tau \le t}, \hat{\theta}_{\tau < t}\right)$, conditioned on the history of neural responses $y_{\tau \le t}$ and the history of point estimates $\hat{\theta}_{\tau < t}$. In the remainder of the Materials and methods, we will formulate all derivations and computations in terms of the history of past estimates (up to and including time $t-1$), with the understanding that these estimates can be used as approximate predictions of the current state at time $t$.

With these simplifications, the general steps of the inference process can be broken down as follows:

1. **Encoder:** maps incoming stimuli $x_{\tau \le t}$ onto a neural response $y_t$ by sampling from the 'encoding distribution' $p\left(y_t | x_{\tau \le t}, \hat{\theta}_{\tau < t}\right)$

2. **Decoder:** uses Bayes' rule to compute the conditional distribution of a stimulus $x_t$ given the neural response $y_t$, which we refer to as the 'decoding distribution' $p\left(x_t | y_t, \hat{\theta}_{\tau < t}\right)$

3. **Observer:** uses the neural response $y_t$ to update the posterior $p\left(\theta_t | y_{\tau \le t}, \hat{\theta}_{\tau < t}\right)$. This can be broken down into the following steps, in which the observer:

   a. Combines the previous posterior $p\left(\theta_{t-1} | y_{\tau < t}, \hat{\theta}_{\tau < t-1}\right)$ with knowledge of environment dynamics $p(\theta_t | \theta_{t-1})$ to compute the probability distribution of $\theta_t$ given all past data, $p\left(\theta_t | y_{\tau < t}, \hat{\theta}_{\tau < t-1}\right)$

   b. Uses Bayes' rule to incorporate a new stimulus $x_t$ and form $p\left(\theta_t | x_t, y_{\tau < t}, \hat{\theta}_{\tau < t-1}\right)$

   c. Marginalizes over the uncertainty in $x_t$ using the decoding distribution $p\left(x_t | y_t, \hat{\theta}_{\tau < t}\right)$, thereby obtaining the updated posterior $p\left(\theta_t | y_{\tau \le t}, \hat{\theta}_{\tau < t}\right)$ (which can be averaged to compute the point estimate $\hat{\theta}_t$)

   d. Combines the updated posterior with knowledge of environment dynamics $p(\theta_{t+1} | \theta_t)$ to generate a predicted distribution of environmental states $p\left(\theta_{t+1} | y_{\tau \le t}, \hat{\theta}_{\tau < t}\right)$ (which can be averaged to compute the point prediction $\vec{\theta}_{t+1}$)

4. **Feedback loop:** sends the prediction back upstream to update the encoder.

In what remains of this section, we derive the general equations for the full inference process in the presence of both encoding and decoding. In Section B, we derive the specific forms of the inference equations in a simplified, two-state environment. We first focus on the general equations of the observer model (Section B.2). We then describe the forms of the encoding and decoding distributions implemented by the three different encoding schemes considered in this paper, and detail how the parameters of each encoder can be optimized based on the observer's prediction of the environmental state (Sections B.3-B.6). In Section C, we describe the numerical approximations used to simulate the results presented in the main paper.

## A.1. Environment dynamics

We consider a non-stationary environment with Markovian dynamics. The dynamics of the environmental state variable $\theta_t$ are then specified by the distribution $p(\theta_t | \theta_{t-1})$. At each time $t$, the value of $\theta_t$ specifies the distribution of stimuli $p(x_t | \theta_t)$.

## A.2. Encoder

We consider an encoder that maps incoming stimuli $x_{\tau \leq t}$ onto a neural response $y_t$. We assume that the encoder has access to the history of estimates $\hat{\theta}_{\tau < t}$ (fed back from a downstream observer) to optimally encode incoming stimuli via the 'encoding distribution', $p\left(y_t | x_{\tau \leq t}, \hat{\theta}_{\tau < t}\right)$.

## A.3. Decoder

Because the observer does not have direct access to the stimulus, it must first decode the stimulus from the neural response. We assume that the decoder has access to the instantaneous neural response $y_t$ and this history of past estimates $\hat{\theta}_{\tau < t}$. The decoder must use these signals to marginalize over past stimuli $x_{\tau < t}$ and compute the probability of the response $y_t$ conditioned on the current stimulus $x_t$ (this probability will later be used to update the observer's posterior):

$$p\left(y_t | x_t, \hat{\theta}_{\tau < t}\right) = \int dx_{\tau < t}\, \underbrace{p\left(y_t | x_t, x_{\tau < t}, \hat{\theta}_{\tau < t}\right)}_{\substack{\text{encoding} \\ \text{distribution}}}\, p\left(x_{\tau < t} | \hat{\theta}_{\tau < t}\right) \tag{5}$$

The decoder must then invert this distribution (using Bayes' rule) to estimate the probability of the stimulus $x_t$ given the response $y_t$ and past estimates $\hat{\theta}_{\tau < t}$:

$$
\begin{aligned}
p\left(x_t | y_t, \hat{\theta}_{\tau < t}\right) &= \frac{p\left(y_t | x_t, \hat{\theta}_{\tau < t}\right) p\left(x_t | \hat{\theta}_{\tau < t}\right)}{p\left(y_t | \hat{\theta}_{\tau < t}\right)} \\
&= \frac{p\left(y_t | x_t, \hat{\theta}_{\tau < t}\right) p\left(x_t | \hat{\theta}_{t-1}\right)}{Z\left(y_t, \hat{\theta}_{\tau < t}\right)}
\end{aligned}
\tag{6}
$$

where we have written the distribution in the denominator as a normalization constant obtained by integrating the numerator:

$$Z\left(y_t, \hat{\theta}_{\tau < t}\right) = \int dx_t\, p\left(y_t | x_t, \hat{\theta}_{\tau < t}\right) p\left(x_t | \hat{\theta}_{t-1}\right) \tag{7}$$

In what follows, we refer to $p\left(x_t | y_t, \hat{\theta}_{\tau < t}\right)$ (defined in *Equation 6*) as the 'decoding distribution'.

## A.4. Observer

The optimal observer should use the decoding distribution to marginalize over its uncertainty about the true value of the stimulus $x_t$ and thereby obtain the posterior probability of $\theta_t$ given past responses $y_{\tau \leq t}$ and past estimates $\hat{\theta}_{\tau < t}$. To do this, we first write an expression for the probability of $\theta_t$ given all data up to (but not including) the current timestep:

$$p\left(\theta_t | y_{\tau < t}, \hat{\theta}_{\tau < t-1}\right) = \int d\theta_{t-1}\, p(\theta_t | \theta_{t-1})\, \underbrace{p\left(\theta_{t-1} | y_{\tau < t}, \hat{\theta}_{\tau < t-1}\right)}_{\substack{\text{posterior from} \\ \text{previous timestep}}} \tag{8}$$

where the prior is taken to be the posterior from the last timestep, and the distribution $p(\theta_t | \theta_{t-1})$ governs the dynamics of the environment.

This distribution can then be combined with a new stimulus $x_t$:

$$p\left(\theta_t|x_t,y_{\tau<t},\hat{\theta}_{\tau<t-1}\right) \;=\; \frac{p\left(x_t|\theta_t,y_{\tau<t},\hat{\theta}_{\tau<t-1}\right)p\left(\theta_t|y_{\tau<t},\hat{\theta}_{\tau<t-1}\right)}{p\left(x_t|y_{\tau<t},\hat{\theta}_{\tau<t-1}\right)}$$

$$=\; \frac{p(x_t|\theta_t)p\left(\theta_t|y_{\tau<t},\hat{\theta}_{\tau<t-1}\right)}{\Omega\left(x_t,y_{\tau<t},\hat{\theta}_{\tau<t-1}\right)}.$$

(9)

As before, we have written the distribution in the denominator as a normalization constant obtained by integrating the numerator:

$$\Omega\left(x_t,y_{\tau<t},\hat{\theta}_{\tau<t-1}\right) = \int d\theta_t\, p(x_t|\theta_t)p\left(\theta_t|y_{\tau<t},\hat{\theta}_{\tau<t-1}\right)$$

(10)

Finally, we marginalize over the unknown value of the signal $x_t$ using the decoding distribution $p\left(x_t|y_t,\hat{\theta}_{\tau<t}\right)$ to obtain the updated posterior distribution:

$$p\left(\theta_t|y_{\tau\le t},\hat{\theta}_{\tau<t}\right) = \int dx_t\, p\left(\theta_t|x_t,y_{\tau<t},\hat{\theta}_{\tau<t-1}\right)p\left(x_t|y_t,\hat{\theta}_{\tau<t}\right)$$

(11)

To form a prediction about the future state of the environment, the observer should combine its belief $p\left(\theta_t|y_{\tau\le t},\hat{\theta}_{\tau<t}\right)$ about the current state of the environment with the knowledge $p(\theta_{t+1}|\theta_t)$ of the environment dynamics in a manner analogous to *Equation 8*.

## A.5. Computing point estimates

The posterior can be used to compute a point estimate $\hat{\theta}_t$ and prediction $\vec{\theta}_{t+1}$ of the environmental state:

$$\hat{\theta}_t = \int d\theta_t\, \theta_t p\left(\theta_t|y_{\tau\le t},\hat{\theta}_{\tau<t}\right) \equiv \hat{\theta}_{y,t}$$

(12)

$$\vec{\theta}_{t+1} \;= \int d\theta_{t+1}\,\theta_{t+1}p\left(\theta_{t+1}|y_{\tau\le t},\hat{\theta}_{\tau<t}\right)$$
$$= \int d\theta_{t+1}\,\theta_{t+1}\int d\theta_t\, p(\theta_{t+1}|\theta_t)p\left(\theta_t|y_{\tau\le t},\hat{\theta}_{\tau<t}\right)$$

(13)

The point estimate given in *Equation 12* is referred to in the main text as '$\hat{\theta}_{y,t}$'. We distinguish this from the point estimate '$\hat{\theta}_{x,t}$', which was derived in *DeWeese and Zador (1998)* in the absence of encoding/decoding.

## B. Model environments

## B.1. Environment dynamics

We consider a two-state environment in which the state $\theta_t$ can take one of two values, $\theta^L$ and $\theta^H$. At each timestep, the environment can switch states with a constant probability $h$, referred to as the 'hazard rate'. The hazard rate fully specifies the dynamics of the environment:

$$\theta_t = z_t\theta_{t-1} + (1-z_t)\left(\theta^L + \theta^H - \theta_{t-1}\right)$$

(14)

where $z_t$ is a binary random variable equal to 1 with probability $h$ and 0 with probability $1-h$.

We take $\theta_t$ to parametrize either the mean $\mu$ or the standard deviation $\sigma$ of a Gaussian stimulus distribution:

$$p(x_t|\theta_t) = \begin{cases} \mathcal{N}(x_t;\theta_t,\sigma^2), & \text{mean-switching environment } (\theta_t = \mu) \\ \mathcal{N}\left(x_t;\mu,\theta_t^2\right), & \text{variance-switching environment } (\theta_t = \sigma) \end{cases}$$

(15)

## B.2. Observer

In a two-state environment, the posterior distribution $p\left(\theta_t|y_{\tau\leq t},\hat{\theta}_{\tau<t}\right)$ can be summarized by a single value $P_t^L = p\left(\theta_t = \theta^L|y_{\tau\leq t},\hat{\theta}_{\tau<t}\right)$, which is the probability that the environment is in the low state at time $t$.

Given the posterior $P_{t-1}^L$ at the previous timestep, the distribution for $\theta_t$ given all past data is given by:

$$p\left(\theta_t = \theta^L|y_{\tau<t},\hat{\theta}_{\tau<t}\right) = (1-h)P_{t-1}^L + h\left(1-P_{t-1}^L\right) \tag{16}$$

where $h$ is the a priori probability that a switch occurred at the current timestep. This distribution can then be combined with a new stimulus $x_t$:

$$\begin{aligned} p\left(\theta_t = \theta^L|x_t,y_{\tau<t},\hat{\theta}_{\tau<t}\right) &= \frac{p(x_t|\theta_t = \theta^L)p\left(\theta_t = \theta^L|y_{\tau<t},\hat{\theta}_{\tau<t}\right)}{\Omega\left(x_t,y_{\tau<t},\hat{\theta}_{\tau<t}\right)} \\ &= \frac{\mathcal{N}\left(x_t;\mu_L,\sigma_L^2\right)\left[(1-h)P_{t-1}^L + h\left(1-P_{t-1}^L\right)\right]}{\Omega\left(x_t,y_{\tau<t},\hat{\theta}_{\tau<t}\right)} \end{aligned} \tag{17}$$

The variables $(\mu_L, \sigma_L)$ and $(\mu_H, \sigma_H)$ correspond to mean and standard deviation of the stimulus distribution in the low and high states, respectively, and their values vary depending on the type of the environment (mean-switching versus variance-switching).

To obtain the updated posterior $P_t^L$, we marginalize over the decoding distribution $p\left(x_t|y_t,\hat{\theta}_{\tau<t}\right)$:

$$\begin{aligned} P_t^L = p\left(\theta_t = \theta^L|y_{\tau\leq t},\hat{\theta}_{\tau<t}\right) &= \int dx_t\, p\left(\theta_t = \theta^L|x_t,y_{\tau<t},\hat{\theta}_{\tau<t}\right)p\left(x_t|y_t,\hat{\theta}_{\tau<t}\right) \\ &= \int dx_t\, \frac{\mathcal{N}\left(x_t;\mu_L,\sigma_L^2\right)\left[(1-h)P_{t-1}^L + h\left(1-P_{t-1}^L\right)\right]}{\Omega\left(x_t,y_{\tau<t},\hat{\theta}_{\tau<t}\right)}p\left(x_t|y_t,\hat{\theta}_{\tau<t}\right) \end{aligned} \tag{18}$$

The posterior can be used to construct a new point-estimate $\hat{\theta}_t$ of the environmental state:

$$\hat{\theta}_t = P_t^L\theta^L + \left(1-P_t^L\right)\theta^H, \tag{19}$$

where $1 - P_t^L = P_t^H$ is the probability that the environment is in the high state at time $t$. Note that although the environmental states are discrete, the optimal Bayesian observer maintains a continuous estimate $\hat{\theta}_t$.

To form a prediction about the future state of the environment, the observer first combines the posterior $P_t^L$ with knowledge of environment dynamics (in a manner analogous to *Equation 16*), and then computes a point prediction (in a manner analogous to *Equation 19*):

$$P_{t+1}^L = P_t^L(1-h) + \left(1-P_t^L\right)h \tag{20}$$

$$\vec{\theta}_{t+1} = P_{t+1}^L\theta^L + \left(1-P_{t+1}^L\right)\theta^H \tag{21}$$

For small hazard rates (as considered here), the predicted value $\vec{\theta}_{t+1}$ is very close to the current estimate $\hat{\theta}_t$. For simplicity, we approximate the prediction $\vec{\theta}_{t+1}$ by the estimate $\hat{\theta}_t$. This estimate is then fed back upstream and used to update the encoder. In the general case, however, one should compute the full predicted distribution of environmental states via *Equation 20*, and use this distribution to optimize the encoder.

## B.3. Encoder/decoder

The posterior (given in *Equation 18*) is a function of the decoding distribution $p\left(x_t|y_t,\hat{\theta}_{\tau<t}\right)$, which depends on the encoding distribution $p\left(y_t|x_{\tau\leq t},\hat{\theta}_{\tau<t}\right)$ through *Equations 5-6*. In what follows, we

derive the encoding and decoding distributions for the three encoding schemes considered in this paper. All three encoding schemes are noiseless; as a result, the encoding distribution $p\left(y_t|x_{\tau \leq t}, \hat{\theta}_{\tau < t}\right)$ reduces to a delta function in each case. This encoding distribution can then be used to derive the decoding distribution, from which it is straightforward to compute the posterior $P_t^L$ via *Equation 18* (and similarly any estimates and predictions derived from the posterior).

Each encoding scheme considered here was parametrized by one or more encoding parameters. In two of the three encoding schemes, these parameters were chosen to minimize an error function $E(x, y)$, subject to a constraint on the fidelity of the encoding. We defined this error function with respect to inference or stimulus reconstruction:

$$E(x, y) = \begin{cases} \left(\hat{\theta}_x - \hat{\theta}_y\right)^2, & \text{error in inference} \\ (x - y)^2, & \text{error in reconstruction} \end{cases} \tag{22}$$

where $\hat{\theta}_y$ was defined in *Equation 12*, and $\hat{\theta}_x$ was derived in *DeWeese and Zador (1998)*.

## B.4. Limited neural response levels: encoding via discretization
### B.4.1. Encoder
Here, we consider a quantization (instantaneous discretization) of the stimulus space that maps the current stimulus $x_t$ onto one of a discrete set of values $\{y_t^i\}$, where $i = 1, 2, \dots N$ labels distinct response levels. This mapping is performed deterministically by choosing the response level that minimizes the instantaneous error $E(x_t, \{y_t^i\})$:

$$\begin{aligned} y_t &= \underset{y_t^i}{\operatorname{argmin}} \, E(x_t, \{y_t^i\}) \\ &= y_t^i \end{aligned} \tag{23}$$

We can therefore write the encoding distribution as a delta function:

$$p\left(y_t|x_{\tau \leq t}, \hat{\theta}_{\tau < t}\right) = \delta\left(x_t - y_t^i\right), \tag{24}$$

where the set of response levels $\{y_t^i\}$ implicitly contains the dependence on $\hat{\theta}_{t-1}$.

### B.4.2. Decoder
The decoder must estimate the probability of a stimulus $x_t$, given that the observed response was $y_t^i$. In principle, the response $y_t^i$ could have been generated by any stimulus in the range $[y_t^{i,L}, y_t^{i,H}]$, where $y_t^{i,L}$ and $y_t^{i,H}$ are the lower and upper bounds of the bin represented by level $y_t^i$, respectively.

The decoding distribution can then be written as a truncated Gaussian distribution:

$$p\left(x_t|y_t, \hat{\theta}_{\tau < t}\right) = \begin{cases} \frac{1}{Z\left(y_t^{i,L}, y_t^{i,H}, \hat{\theta}_{t-1}\right)} \mathcal{N}\left(x_t; \hat{\mu}_{t-1}, \hat{\sigma}_{t-1}^2\right), & y_t^{i,L} < x_t < y_t^{i,H} \\ 0, & \text{otherwise} \end{cases} \tag{25}$$

where $Z\left(y_t^{i,L}, y_t^{i,H}, \hat{\theta}_{t-1}\right)$ is a normalization constant. For simplicity, we approximated this truncated Gaussian distribution with a delta function:

$$p\left(x_t|y_t, \hat{\theta}_{\tau < t}\right) \approx \delta\left(x_t - y_t^i\right) \tag{26}$$

We verified numerically that this approximation did not impact our results.

### B.4.3. Determining the optimal response levels
At each point in time, the optimal set of response levels $\{y_t^i\}^*$ was found by minimizing the following equation:

$$\{y_t^i\}^* = \underset{\{y_t^i\}}{\operatorname{argmin}} \, \left\langle E\left(x_t, \{y_t^i\}\right)\right\rangle_{p\left(x_t|\hat{\theta}_{t-1}\right)} \tag{27}$$

subject to a hard constraint on the number of response levels. When optimizing for mean-switching environments, we defined the error function with respect to the raw stimulus and neural response (i.e. $E = E(x,y)$). When optimizing for variance-switching environment, we defined the error function with respect to the absolute value of the stimulus and neural response (i.e. $E = E(|x|,|y|)$). We computed $\langle E(x_t, \{y_t^i\}) \rangle$ numerically; see Section C.3.1.

## B.5. Limited gain and temporal acuity: encoding via temporal filtering

### B.5.1. Encoder

In this encoding scheme, we consider a simple temporal filter parameterized by the coefficient $\alpha_t$. This filter linearly combines current ($x_t$) and past ($x_{t-1}$) stimuli:

$$y_t = \alpha_t x_t + (1 - \alpha_t) x_{t-1} \tag{28}$$

The encoding distribution is then given by:

$$p\left(y_t | x_{\tau \leq t}, \hat{\theta}_{\tau < t}\right) = \delta(y_t - (\alpha_t x_t + (1 - \alpha_t) x_{t-1})), \tag{29}$$

where the filtering coefficient $\alpha_t$ implicitly contains the dependence on $\hat{\theta}_{t-1}$.

### B.5.2. Decoder

The encoding is a function of both current and past stimuli. The decoder, however, only has access to the current response $y_t$. In order to estimate the probability that this response was generated by the stimulus $x_t$, the decoder must first use the internal estimates $\hat{\theta}_{\tau < t}$ to marginalize over uncertainty in past stimuli $x_{\tau < t}$. This was first outlined in **Equation 5**, which reduces here to:

$$
\begin{aligned}
p\left(y_t | x_t, \hat{\theta}_{\tau < t}\right) &= \int dx_{\tau < t}\, p\left(y_t | x_t, x_{\tau < t}, \hat{\theta}_{\tau < t}\right) p\left(x_{\tau < t} | \hat{\theta}_{\tau < t}\right) \\
&= \int dx_{t-1}\, p\left(y_t | x_t, x_{t-1}, \hat{\theta}_{t-1}\right) p\left(x_{t-1} | \hat{\theta}_{t-1}\right) \\
&= \int dx_{t-1}\, \delta(y_t - (\alpha_t x_t + (1 - \alpha_t) x_{t-1})) \mathcal{N}\left(x_{t-1}; \hat{\mu}_{t-1}, \hat{\sigma}_{t-1}^2\right) \\
&= \frac{1}{(1 - \alpha_t)} \mathcal{N}\left(\frac{y_t - \alpha_t x_t}{(1 - \alpha_t)}; \hat{\mu}_{t-1}, \hat{\sigma}_{t-1}^2\right) \\
&= \mathcal{N}\left(y_t; \alpha_t x_t + (1 - \alpha_t)\hat{\mu}_{t-1}, (1 - \alpha_t)^2 \hat{\sigma}_{t-1}^2\right)
\end{aligned}
\tag{30}
$$

The decoder can then use Bayes' rule to invert this distribution and determine the probability of the stimulus $x_t$ given the response $y_t$:

$$
\begin{aligned}
p\left(x_t | y_t, \hat{\theta}_{\tau < t}\right) &= \frac{p\left(y_t | x_t, \hat{\theta}_{\tau < t}\right) p\left(x_t | \hat{\theta}_{\tau < t}\right)}{Z\left(y_t, \hat{\theta}_{\tau < t}\right)} \\
&= \frac{1}{Z\left(y_t, \hat{\theta}_{\tau < t}\right)} \mathcal{N}\left(y_t; \alpha_t x_t + (1 - \alpha_t)\hat{\mu}_{t-1}, (1 - \alpha_t)^2 \hat{\sigma}_{t-1}^2\right) \mathcal{N}\left(x_t; \hat{\mu}_{t-1}, \hat{\sigma}_{t-1}^2\right)
\end{aligned}
\tag{31}
$$

In its current form, this decoding distribution is written as a Gaussian over the variable $y_t$. Ultimately, the observer must use this decoding distribution to marginalize over uncertainty in $x_t$. In Appendix I, we walk through the algebra needed to rewrite this distribution as Gaussian over $x_t$. The final form of this distribution in given by:

$$p\left(x_t | y_t, \hat{\theta}_{\tau < t}\right) = \mathcal{N}\left(x_t; \frac{\alpha_t y_t - (1 - \alpha_t)(2\alpha_t - 1)\hat{\mu}_{t-1}}{1 - 2\alpha_t + 2\alpha_t^2}, \left(\frac{(1 - \alpha_t)^2}{1 - 2\alpha_t + 2\alpha_t^2}\right)\hat{\sigma}_{t-1}^2\right) \tag{32}$$

### B.5.3. Determining the optimal filter coefficient

The optimal filtering coefficient $\alpha_t^*$ was found by minimizing the following equation:

$$\alpha_t^* = \underset{\alpha_t}{\operatorname{argmin}}\ \langle E(x_t, y_t) \rangle_{p\left(x_t | \hat{\theta}_{t-1}\right)} + \beta H\left(y_t, y_{t+1} | \hat{\theta}_{\tau < t}\right) \tag{33}$$

The error term, $\langle E(x_t, y_t) \rangle$, was computed numerically; see Section C.3.2. The entropy term, $H\left(y_t, y_{t+1} | \hat{\theta}_{\tau < t}\right)$, can be computed analytically (see Appendix 2 for details):

$$
\begin{aligned}
H\left(y_t, y_{t+1} | \hat{\theta}_{\tau < t}\right) &= H\left(y_{t+1} | y_t, \hat{\theta}_{\tau < t}\right) + H\left(y_t | \hat{\theta}_{\tau < t}\right) \\
&= \frac{1}{2} \log\left(4\pi^2 e^2 \left(\alpha_t^2 \hat{\sigma}_{t-1}^2 + \frac{(1-\alpha_t)^4}{1 - 2\alpha_t + 2\alpha_t^2} \hat{\sigma}_{t-1}^2\right)\left(\alpha_t^2 \hat{\sigma}_{t-1}^2 + (1-\alpha_t)^2 \hat{\sigma}_{t-1}^2\right)\right)
\end{aligned}
\tag{34}
$$

## B.6. Limited neural activity: encoding via dynamic stimulus selection

### B.6.1. Encoder

In this encoding scheme, the encoder uses the misalignment signal $M_t$ to determine whether or not to encode and transmit the stimulus $x_t$. If the magnitude of the misalignment signal exceeds the threshold $V$, the stimulus is encoded and transmitted. Otherwise, the stimulus is not encoded, and a 'null symbol' is transmitted to the observer. For the purposes of computing the encoding and decoding distributions, we use $y_t = 0$ to denote the null symbol (in the main text, we denoted the null symbol by $y_t = \emptyset$).

This encoding is a deterministic mapping of the stimulus $x_t$ onto the response $y_t$, dependent upon the misalignment signal $M_t$. The encoding distribution can thus be written in a probabilistic form as a mixture of two delta functions:

$$
p\left(y_t | x_t, \hat{\theta}_{\tau < t}\right) = \begin{cases} \delta(y_t - x_t), & |M_t| > V \\ \delta(y_t), & |M_t| \leq V \end{cases}
\tag{35}
$$

where $M_t$ implicitly contains the dependence on $\hat{\theta}_{t-1}$.

### B.6.2. Decoder

In this scheme, the form of the decoding distribution depends on whether or not the encoder transmits the stimulus $x_t$. If the stimulus was encoded and transmitted, there is no uncertainty in its value, and the decoding distribution is a delta function about $y_t$. If the stimulus was not encoded and the null symbol was instead transmitted, the decoder can only assume that the stimulus came from the estimated stimulus distribution $p\left(x_t | \hat{\theta}_{t-1}\right)$.

The decoding distribution therefore takes the following form:

$$
p\left(x_t | y_t, \hat{\theta}_{\tau < t}\right) = \begin{cases} \delta(x_t - y_t), & y_t \neq 0 \\ p\left(x_t | \hat{\theta}_{t-1}\right), & y_t = 0 \end{cases}
\tag{36}
$$

### B.6.3. Determining the misalignment signal

In defining this encoding scheme, our aim was to construct a heuristic 'misalignment' signal that would alert the encoder to a change in the stimulus distribution. One candidate is a signal that tracks the average surprise of incoming stimuli, given the internal estimate of the environmental state.

The surprise associated with a single stimulus $x_t$ is equal to the negative log probability of the stimulus given the estimate $\hat{\theta}_{t-1}$:

$$
S(x_t) = -\log\left[p\left(x_t | \hat{\theta}_{t-1}\right)\right]
\tag{37}
$$

The average surprise of incoming stimuli, obtained by averaging over the true stimulus distribution $p(x_t | \theta_t)$, is equal to cross-entropy between the true and estimated stimulus distributions:

$$
H\left(x_t; \theta_t, \hat{\theta}_{t-1}\right) = \int dx_t \, S(x_t) p(x_t | \theta_t)
\tag{38}
$$

$$
= H(x_t; \theta_t) + D_{KL}\left[p(x_t | \theta_t) \| p\left(x_t | \hat{\theta}_{t-1}\right)\right],
\tag{39}
$$

where the second term in *Equation 39* is the Kullback-Leibler divergence of the estimated stimulus distribution from the true stimulus distribution.

The cross-entropy is equal to the entropy of the true stimulus distribution when the observer's estimate is accurate (i.e., when $\hat{\theta}_{t-1} = \theta_t$), and increases with the inaccuracy of the observer's estimate. To construct a signal that deviates from zero (rather than from the entropy of the stimulus distribution) whenever observer's estimate is inaccurate, we subtract the estimated entropy $H\left(x_t; \hat{\theta}_{t-1}\right)$ from the cross-entropy to define the 'misalignment signal':

$$M_t = H\left(x_t; \theta_t, \hat{\theta}_{t-1}\right) - H\left(x_t; \hat{\theta}_{t-1}\right) \tag{40}$$

$$= H(x_t; \theta_t) + D_{KL}\left[p(x_t|\theta_t) \| p\left(x_t|\hat{\theta}_{t-1}\right)\right] - H\left(x_t; \hat{\theta}_{t-1}\right) \tag{41}$$

The magnitude of this signal is large whenever the average surprise of incoming stimuli differs from the estimated surprise, and monotonically increases as a function of the difference between the true and estimated states of the environment. In the case of a Gaussian distribution, the misalignment signal reduces to:

$$M_t = \frac{1}{2}\log\left(2\pi e\sigma_t^2\right) + \left(\log\left(\frac{\hat{\sigma}_{t-1}}{\sigma_t}\right) + \frac{\sigma_t^2 + (\mu_t - \hat{\mu}_{t-1})^2}{2\hat{\sigma}_t^2} - \frac{1}{2}\right) - \frac{1}{2}\log\left(2\pi e\hat{\sigma}_{t-1}^2\right) \tag{42}$$

where $\mu_t$ and $\sigma_t$ are the mean and standard deviation of the true stimulus distribution, respectively, and $\hat{\mu}_{t-1}$ and $\hat{\sigma}_{t-1}$ are the estimated values of the same parameters. The analytical values of this misalignment signal are plotted in *Figure 2I–J*.

In practice, the encoder does not have access to the parameters of the true stimulus distribution, and must therefore estimate the misalignment signal directly from incoming stimulus samples. This is discussed in more detail in Section C.3.3.

## C. Numerical simulations

### C.1. Environment parameters

All results were generated using a probe environment in which the state $\theta_t$ switched between two fixed values, $\theta^L$ and $\theta^H$, every 100 time samples (corresponding to a hazard rate of $h = 0.01$). A single cycle of this probe environment consists of 200 time samples, for which the environment is in the low state ($\theta_t = \theta^L$) for the first 100 time samples and in the high state ($\theta_t = \theta^H$) for the second 100 time samples. In the main text, we averaged results over multiple cycles of the probe environment.

For the mean-switching environment, the state $\theta_t$ parametrized the mean of the stimulus distribution and switched between $\mu = \theta^L = -1$ and $\mu = \theta^H = 1$. The standard deviation was fixed to $\sigma = 1$. For the variance-switching environment, $\theta_t$ parametrized the standard deviation of the stimulus distribution and switched between $\sigma = \theta^L = 1$ and $\sigma = \theta^H = 2$. The mean was fixed to $\mu = 0$.

### C.2. Updating the posterior

On each timestep, a single stimulus $x_t$ was drawn randomly from $p(x_t|\theta_t)$. The stimulus was encoded, decoded, and used to update the posterior $P_t^L$. Updating the posterior requires marginalizing over the decoding distribution $p\left(x_t|y_t, \hat{\theta}_{\tau<t}\right)$ (as given by *Equation 11*). We approximated this marginalization numerically via Monte-Carlo simulation. At each time step, we generated 200 samples from the decoding distribution specified by each encoding scheme (for reference, the decoding distributions are given in *Equations 26, 32, and 36*). Individual samples were then used to compute separate estimates of the posterior, and the resulting set of estimates was averaged over samples. Results were robust to the number of samples used, provided that this number exceeded 50. In the case of encoding via discretization, we found that results were not sensitive to the inclusion of this marginalization step. We therefore computed all results for the discretization encoding scheme in the absence of marginalization by using the neural response $y_t$ to directly update the posterior. This posterior forms the basis of all estimates $\hat{\theta}_t$ and predictions $\vec{\theta}_{t+1}$.

### C.3. Optimizing the encoding

For two of the three encoding schemes (discretization and temporal filtering), the estimate $\hat{\theta}_{t-1}$ was used to optimize a set of encoding parameters (the set of neural response levels $\{y_t^i\}$ in the case of

discretization, and the filtering coefficient $\alpha_t$ in the case of temporal filtering). To perform these optimizations, we discretized the posterior $P_t^L$ into 100 values equally spaced between 0 and 1. This resulted in a set of 100 discretized values of the estimated state $\hat{\theta}_{bin}$. We found the optimal encoding parameters for each value of $\hat{\theta}_{bin}$ (described in detail in the following sections); this resulted in 100 sets of optimal response levels (given a fixed number of levels), and 100 values of the filtering coefficient $\alpha$ (given a fixed constraint strength $\beta$). On each timestep of the simulation, the true estimate $\hat{\theta}_t$ was mapped onto the closest discretized value $\hat{\theta}_{bin}$. The corresponding encoding parameters were then used to encode the incoming stimulus $x_t$. Additional details of each optimization procedure are described in the following sections.

## C.3.1. Limited neural response levels: encoding via discretization

Response levels were chosen to optimize the following objective function:

$$\left\{y_t^i\right\}^* = \underset{\{y_t^i\}}{\mathrm{argmin}} \left\langle E\left(x_t, \{y_t^i\}\right)\right\rangle_{p\left(x_t|\hat{\theta}_{t-1}\right)} \tag{43}$$

The optimal set of response levels $\left\{y_t^i\right\}^*$ was found numerically using Lloyd's algorithm (**Cover and Thomas, 2012**) (often referred to as K-means clustering). The algorithm takes the following as inputs: a set of points to be clustered $\{x\}$ (corresponding to stimulus samples), a number of quantization levels $N$ (corresponding to the number of neural response levels), and a distortion measure $d(x, y)$ (corresponding to the error function $E(x, y)$). The goal of the algorithm is to find a quantization (what we referred to as a discretization of the stimulus space) that minimizes the average value of the distortion.

The values of the quantization levels, $y^1, \ldots, y^N$, are first randomly initialized. The algorithm then proceeds in two steps:

1. Each point $x$ is assigned to a quantization level $y^i$ that yields the smallest distortion $d(x, y^i)$.
2. Each quantization level is replaced by the average value of the points assigned to it.

The two steps are iterated until convergence.

We computed a set of optimal quantization levels (optimal response levels) for each of the 100 discretized values of $\hat{\theta}_{bin}$ (described above). For each value of $\hat{\theta}_{bin}$, we generated a training dataset $\{x\}$ consisting of 50,000 values drawn from the estimated stimulus distribution $p\left(x_t|\hat{\theta}_{bin}\right)$. We determined the boundaries of each quantization level (i.e., the values $y^{i,L}$ and $y^{i,H}$ that bounded the set of stimuli that were mapped to the same quantization level) by assigning points in the training dataset to the quantization level $y^i$ that minimized $d(x, y^i)$.

To compute optimal quantization levels for stimulus reconstruction, we used the standard distortion measure $d(x, y) = (x - y)^2$; in this case, the algorithm is guaranteed to converge to the global optimum. To compute optimal quantization levels for inference, we defined the distortion measure to be $d(x, y) = \left(\hat{\theta}_x - \hat{\theta}_y\right)^2$. The algorithm is not guaranteed to converge to a global optimum in this case, but we found empirically that the algorithm converged to a local optimum (**Figure 3—figure supplement 1**). Moreover, the two distortion measures did not produce equivalent results.

## C.3.2. Limited gain and temporal acuity: encoding via temporal filtering

The optimal filtering coefficient was chosen to minimize the following objective function:

$$\alpha_t^* = \underset{\alpha_t}{\mathrm{argmin}} \left\langle E(x_t, y_t)\right\rangle_{p\left(x_t|\hat{\theta}_{t-1}\right)} + \beta H\left(y_t, y_{t+1}|\hat{\theta}_{\tau<t}\right), \tag{44}$$

where as before, we choose $E(x, y) = \left(\hat{\theta}_x - \hat{\theta}_y\right)^2$ when optimizing for inference, and $E(x, y) = (x - y)^2$ when optimizing for reconstruction.

The joint entropy $H\left(y_t, y_{t+1}|\hat{\theta}_{\tau<t}\right)$ can be determined analytically, as derived in Section B.5.3. We approximated the error term, $\left\langle E(x_t, y_t)\right\rangle_{p\left(x_t|\hat{\theta}_{t-1}\right)}$, numerically. To do so, we first discretized $\alpha$ into 50 values evenly spaced between 0 and 1 (corresponding to 50 discrete values of $\alpha_{bin}$). As described

above, we also discretized the posterior $P_t^L$ into 100 values (corresponding to 100 discrete values of $\hat{\theta}_{bin}$). For each combination of $\alpha_{bin}$ and $\hat{\theta}_{bin}$, we generated 50,000 pairs of stimulus samples $(x_{t-1}, x_t)$ from the distribution $p\left(x_t|\hat{\theta}_{t-1}\right)$. Each sample was used to compute values of the estimates $\hat{\theta}_x$ and $\hat{\theta}_y$. The errors $\left(\hat{\theta}_x - \hat{\theta}_y\right)^2$ and $(x_t - y_t)^2$ were then averaged over all 50,000 stimulus pairs.

The optimal value $\alpha_t^*$ was then chosen as the value of $\alpha_{bin}$ that minimized the objective in *Equation 44* for a given choice of the error function $E(x, y)$ and constraint strength $\beta$.

## C.3.3. Limited neural activity: encoding via dynamic stimulus selection

The misalignment signal, derived in Section B.6.3, was defined in terms of the relative alignment between the true stimulus distribution, $p(x_t|\theta_t)$, and the estimated stimulus distribution, $p\left(x_t|\hat{\theta}_{t-1}\right)$. When the parameters of the true stimulus distribution are known, the value of this signal can be computed analytically via *Equation 40*. However, when the system does not have access to the stimulus distribution (as is the case here), this signal must be estimated directly from incoming stimulus samples. We consider a scenario in which the encoder can approximate *Equation 40* by computing a running-average of the stimulus surprise:

$$M_t = -\frac{1}{T}\sum_{\tau=0}^{T} \log\left[p\left(x_{t-\tau}|\hat{\theta}_{t-1}\right)\right] - H\left(x_t|\hat{\theta}_{t-1}\right),$$  (45)

where $T$ specifies the number of timebins used to estimate the average surprise. All results in the main text were generated using $T = 10$ timebins.

## C.4. The role of surprise and uncertainty

*Figure 1B–D* illustrated the relative impact of different stimuli on the observer's estimate of an environmental state $\theta$, which is modulated by the observer's uncertainty and the surprise of incoming stimuli (for numerical values of the color ranges in *Figure 1B–D*, see *Figure 1—figure supplement 2*).

To illustrate this, we considered optimal Bayesian estimation of the location $\mu$ and scale $\alpha^2/2$ of a generalized Gaussian stimulus distribution:

$$p(x; \mu, \alpha, \beta) = \frac{\beta}{2\alpha\Gamma(1/\beta)}\exp\left[-\left(\frac{|x-\mu|}{\alpha}\right)^\beta\right]$$  (46)

Our derivation is analogous to that outlined in *Murphy (2007)* for estimating the mean of a Gaussian stimulus distribution.

We consider a snapshot of the inference process, when the observer's prior is centered around a fixed estimate of the location ($\hat{\mu} = 0$) or scale ($\hat{\alpha}^2/2 = 1$). When estimating location, we fix the scale parameter to be $\alpha = \sqrt{2}$ (corresponding to a Gaussian distribution with variance $\sigma^2 = \alpha^2/2 = 1$ when $\beta = 2$). When estimating scale, we fix the location parameter to be $\mu = 0$. In both cases, we consider three different values of the shape parameter: $\beta = 1, 2, 10$.

The surprise of a single stimulus observation is quantified by the negative log probability of the stimulus value given the observer's estimate. We consider 100 evenly-spaced values of surprise between 1 and 10. For each value of surprise, we compute the value of the stimulus $x_t^*$ that elicits a given surprise.

The observer's uncertainty is captured by the entropy of the prior distribution. When estimating the location parameter, the natural conjugate prior is the Gaussian distribution $\mathcal{N}\left(\mu; \mu_0, \sigma_0^2\right)$ with mean $\mu_0 = \hat{\mu}$ (we take this mean to be the observer's point estimate of the environmental state before observing a stimulus sample $x_t^*$, that is, $\hat{\theta}_{t-1} = \hat{\mu}$). The entropy of the prior distribution depends only on its variance: $H = \frac{1}{2}\log\left(2\pi e\sigma_0^2\right)$. We consider 100 evenly-spaced values of the entropy between 0 and 0.7. For each value of entropy, we compute the value $\sigma_0^2 = \exp(2H)/2\pi e$ that elicits a given entropy.

When estimating the scale parameter, the natural conjugate prior is the inverse gamma function $p(\alpha; \alpha_0, \beta_0)$ with mean $\hat{\alpha} = \beta_0/(\alpha_0 - 1)$ (we take $\hat{\theta}_{t-1} = \hat{\alpha}^2/2$ to be the observer's estimate of

the environmental state before observing $x_t^\star$). The entropy of the prior depends on both $\alpha_0$ and $\beta_0$: $H = \alpha_0 + \log(\beta_0\Gamma(\alpha_0)) - (1+\alpha_0)\Psi(\alpha_0)$. We fix $\beta_0 = \hat{\alpha}(\alpha_0 - 1)$. We note that the entropy is non-monotonic in $\alpha_0$; we restrict our analysis to values $\alpha_0 > 2$ where both the mean and the variance of the prior are well-defined, and the entropy is monotonic. We again consider 100 evenly-spaced values of the entropy between 0 and 0.7. For each value of entropy, we compute the value $\alpha_0$ that elicits a given entropy.

For each combination of prior uncertainty and surprise, we computed the posterior either over the location parameter, or over the scale parameter. We then computed the squared difference between the average value of the prior and the average value of the posterior $\left((\hat{\mu}_{t-1} - \hat{\mu}_t(x_t^\star))\right)^2$ in the case of location estimation, and $\left(\hat{\alpha}_{t-1}^2/2 - \hat{\alpha}_t(x_t^\star)^2/2\right)^2$ in the case of scale estimation), and we used this squared difference as a measure of the impact of a single stimulus observation on the observer's estimate of location or scale. When reporting the results in *Figure 1B–D*, we separately scaled heatmaps for each stimulus distribution (Laplace, Gaussian, and flat) and for each estimated parameter (location and scale); numerical ranges of these heatmaps are given in *Figure 1—figure supplement 2*.

## C.5. Generating spike rasters

*Figure 6A* showed simulated spike rasters for an encoding scheme with limited neural response levels. To generate these rasters, a stimulus sample $x_t$ was randomly drawn from the true stimulus distribution $p(x_t|\theta_t)$. This stimulus was then mapped onto one of $N=4$ neural response levels. Each response level was assigned a binary spike pattern from the set $\{[00], [10], [01], [11]\}$, where 1 or 0 correspond to presence or absence of a spike, respectively. Patterns were assigned to response levels $\{y_t^i\}$ according to the probability $p(y_t^i|\theta_t)$ that a particular level would be used to encode incoming stimuli. In this way, the pattern with the fewest spikes ($[00]$) was assigned to the response level with the highest probability, and the pattern with the most spikes ($[11]$) was assigned to the level with the lowest probability. This strategy (called 'entropy coding') achieves the shortest average encoding of the input by using the fewest number of spikes (*Cover and Thomas, 2012*). We simulated spike patterns for 800 cycles of the probe environment using the set of response levels optimized for inference or stimulus reconstruction.

## C.6. Computing metamer probabilities

We estimated the probability of a metamer as a function of the alignment between the true state of the environment $\theta$ and the observer's prediction $\vec{\theta}$. We say that two stimuli $x_t^1$ and $x_t^2$ are metamers (i.e., they are indistinguishable to the observer) if in the process of encoding they become mapped on the same neural response level $y^M$ (i.e., $y_t^1 = y_t^2 = y^M$). The probability of a metamer, $p\left(y_t^1 = y_t^2|\theta_t, \hat{\theta}_{t-1}\right)$, depends on both the true and predicted states of the environment. We numerically estimated this probability for a mean-switching environment in the low state ($\theta = \theta^L$). We generated 100 values of $\hat{\theta}_{t-1}$, evenly spaced between $\theta^L$ and $\theta^H$. For each value of $\hat{\theta}_{t-1}$, we drew 100,000 pairs of samples from the stimulus distribution $p(x_t|\theta_t = \theta^L)$. We encoded each stimulus by mapping it onto the corresponding response level $y_t$ (using an encoder with eight response levels, optimized as described in Section C.3.1). If both stimuli in the pair were mapped on the same response level, we counted the trial as a metamer. The total probability of a metamer was computed as the proportion of all trials that resulted in metamers.

## C.7. The role of transmission noise

To better understand the influence of noise on the inference process, we analyzed the behavior of the discretization encoding scheme in the presence of noise. Gaussian noise with variance $\sigma_n^2$ was added to the response $y_t$ of the encoder prior to computing the estimate $\hat{\theta}_t$ (*Figure 3—figure supplement 2A–B*). This form of noise can be viewed as neuronal noise introduced in the transmission of the stimulus representation to downstream areas. The performance of the optimal observer (*Figure 3—figure supplement 2C*) was relatively robust at low noise levels (up to $\sigma_n^2 = 0.4$), but

decreased substantially at high noise levels. A more thorough investigation of the role of noise on optimal inference and encoding strategies is a subject of future work.

## C.8. Measuring speed and accuracy of inference

*Figure 6D-E* compared the accuracy and speed of inference across different encoding schemes and environments. Accuracy was computed separately for the high and low states ($\theta = \theta^H$ and $\theta = \theta^L$, respectively) using the posterior $P_t^L$. For each time point, we first computed the average value of $P_t^L$ across many cycles of the probe environment (500 cycles for discretization, and 800 cycles for filtering and thresholding, corresponding to the average trajectories of $\hat{\theta}$ shown in *Figures 3–5*).

If the observer's estimate is accurate, $P_t^L$ should be close to one when the environment is in the low state, and $(1 - P_t^L)$ should be close to one when the environment is in the high state. We therefore computed the time-averaged values $\langle P_t^L \rangle_t$ and $\langle (1 - P_t^L) \rangle_t$ to measure the accuracy in the low and high states, respectively. Time-averages were computed over the final 10 timesteps in the high or low state, respectively, corresponding to the adapted portion of the inference process.

Speed was computed separately for downward versus upward switches in the environment by measuring the number of time samples required for the posterior to stabilize after a switch. We used the time-averaged value $\langle P_t^L \rangle_t$ (again averaged over the final 10 timesteps) as a measure of the final value of the posterior in the low state. We then counted the number of timesteps after a switch downward before the posterior came within 0.05 of this value, and we used the inverse of this time as a measure of speed. We computed the speed of response to an upward switch in an analogous manner.

## C.9. Natural image simulation

*Figure 7* illustrated a model visual inference task performed on natural images. Within this model, the encoder implemented a sparse encoding of individual image patches (*Olshausen and Field, 1996*) using 32 basis functions $\phi_i$. The basis functions were chosen to minimize the following cost function:

$$\sum_n \left( x_{t,n} - \sum_i y_{i,t} \phi_{i,n} \right)^2 + \lambda \sum_i |y_{i,t}|, \tag{47}$$

where $\vec{x}_t$ is an image patch, $\vec{y}_t$ is the neural population response, $n$ indexes pixels in an image patch, and $i$ indexes neurons in the population. The first term imposes a cost on reconstruction error between the image patch $\vec{x}_t$ and the reconstructed patch $\hat{x}_t = \sum y_{i,t} \phi_i$. The second term imposes a penalty for large population responses. The parameter $\lambda$ imposes a constraint on the fidelity of the encoding by controlling the overall sparsity of the population response.

The set of basis functions was trained on 50,000 image patches of size $16 \times 16$ pixels. Image patches were drawn randomly from the van Hateren database (*van Hateren and Ruderman, 1998*). During training, the value of the sparsity parameter $\lambda$ was set to 0.1.

A sparse representation $\vec{y}_t$ was inferred for each image patch $\vec{x}_t$ via gradient-descent on the cost function in *Equation 47* (*Olshausen and Field, 1996*). An image reconstruction $\hat{x}_t$ was computed from the sparse representation (*Figure 7—figure supplement 1A*). The reconstructed patch was contrast normalized by dividing each pixel value by the standard deviation across the set of pixel values. The normalized image patch was projected onto four curvature filters $C_j$, resulting in four curvature coefficients $v_{j,t}$. Curvature filters were hand-designed to bear coarse, qualitative resemblance to curvature-selective receptive fields in V2. The set of four curvature coefficients was used to update the posterior distribution over variance, analogous to the Bayesian estimation of variance described in Section C.4.

Image areas 1 (low curvature) and 2 (high curvature) in *Figure 7* were chosen to be $200 \times 200$ pixels in size. For illustrative purposes, they were selected to generate a relatively large difference in the variance of curvature filters, which would require a substantial update of the Bayesian estimate. During all simulations, the mean of the prior (corresponding to the observer's point estimate $\hat{\theta}_{t-1}$) was fixed to 5.3, equal to the variance of filter outputs in image area 1.

To numerically compute the impact of a stimulus on the estimate as a function of observer's uncertainty (prior variance) and centered surprise (*Figure 7D*), a set of 5,000 image patches was drawn randomly from image area 2. Image patches were then sorted according to their centered surprise and divided into 5 groups that uniformly spanned the range of centered surprise in the set. The variance of the prior was chosen to be one of 5 equally spaced values between 0.018 and 0.18. For each value of prior variance and for each group of stimuli with a given centered surprise, we computed the change in the observer's estimate before and after incorporating the population response $\vec{y}_t$: $\left( \hat{\theta}_{t-1} - \hat{\theta}_t(\vec{y}_t) \right)^2$.

We used a similar approach to numerically compute the inference error as a function of the sparsity parameter $\lambda$ and the centered surprise (*Figure 7F*). We chose 5 equally-spaced values of $\lambda$ between 0.1 and 10. We then randomly drew 5,000 image patches from image area 2. Image patches were again sorted according to their centered surprise and were divided into 5 groups that uniformly spanned the range of centered surprise in the set. We then computed the average inference error for each value of $\lambda$ and each stimulus group. An analogous procedure was used to determine the inference error as a function of the sparsity parameter $\lambda$ and the observer's uncertainty (*Figure 7E*).

The abrupt changes in impact and inference error that can be seen in *Figure 7D–F* are a result of the coarse partitioning of the set of image patches into a small number of groups. In comparison, the results in *Figure 1B–D* were computed analytically with continuous values of surprise and uncertainty, and therefore show smooth variations in impact and error.

Simulated population responses (*Figure 7E–F*) were generated by selecting a random subset of 45 image patches with a given centered surprise, or specified values of uncertainty. Image patches were then encoded with a sparsity value of either $\lambda = 0.1$ or $\lambda = 10$ (corresponding to the inference errors marked with red and white circles). 40 images patches were encoded with the higher value of $\lambda$, and 5 image patches were encoded with the lower value of $\lambda$. For illustrative purposes, the image patches were arranged such that the first and last 20 patches corresponded to high values of $\lambda$ values (white), while the middle 5 patches correspond to low values of $\lambda$ (red). High and low values of $\lambda$ were chosen to generate similar average inference error for the given values of centered surprise and uncertainty.

Centered surprise was computed for each image patch $\vec{x}_t$ as follows:

$$\sum_j \left| H\left[ p\left( v_{j,t} | \hat{\theta}_{t-1} \right) \right] + \log\left[ p\left( v_{j,t} | \hat{\theta}_{t-1} \right) \right] \right| \tag{48}$$

where $H\left( v_{j,t} | \hat{\theta}_{t-1} \right) = \frac{1}{2}\log\left( 2\pi e \hat{\theta}_{t-1}^2 \right)$ is the entropy of the Gaussian distribution of curvature coefficients given the prior estimate $\hat{\theta}_{t-1}$.

## Acknowledgements

We thank John Briguglio, Vivek Jayaraman, Yarden Katz, Emily Mackevicius, and Josh McDermott for useful discussions and feedback on the manuscript. WM was supported by the Center for Brains, Minds and Machines (CBMM), funded by NSF STC award CCF-1231216. AMH was supported by the Howard Hughes Medical Institute.

## Additional information

### Funding

| Funder | Grant reference number | Author |
| --- | --- | --- |
| National Science Foundation | STC Award CCF-1231216 | Wiktor F Mlynarski |
| Howard Hughes Medical Institute | | Ann M Hermundstad |

The funders had no role in study design, data collection and interpretation, or the decision to submit the work for publication.

## Author contributions
Wiktor F Młynarski, Ann M Hermundstad, Conceptualization, Software, Formal analysis, Investigation, Visualization, Methodology, Writing—original draft, Writing—review and editing

## Author ORCIDs
Wiktor F Młynarski http://orcid.org/0000-0002-3791-5656
Ann M Hermundstad http://orcid.org/0000-0002-0377-0516

## Decision letter and Author response
Decision letter https://doi.org/10.7554/eLife.32055.020
Author response https://doi.org/10.7554/eLife.32055.021

# Additional files
## Supplementary files
• Transparent reporting form
DOI: https://doi.org/10.7554/eLife.32055.016

## Data availability
All data generated or analysed during this study are included in the manuscript and supporting files.

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

## Appendix 1

DOI: https://doi.org/10.7554/eLife.32055.017

Here, we provide a detailed derivation of the decoding distribution for the filtering encoder (described in Section B.5.2).

To simplify *Equation 31*, we rewrite the first Gaussian as a function of $\alpha_t x_t$ (for notational simplicity, we will write $\sigma'^2 = (1-\alpha_t)^2 \hat{\sigma}_{t-1}^2$:

$$
\begin{aligned}
\mathcal{N}(\star) &= \mathcal{N}(y_t; \alpha_t x_t + (1-\alpha_t)\hat{\mu}_{t-1}, \sigma'^2) \\
&= \frac{1}{\sqrt{2\pi\sigma'^2}} \exp\left(-\frac{(y_t - (\alpha_t x_t + (1-\alpha_t)\hat{\mu}_{t-1}))^2}{2\sigma'^2}\right) \\
&= \frac{1}{\sqrt{2\pi\sigma'^2}} \exp\left(-\frac{(\alpha_t x_t - (y_t - (1-\alpha_t)\hat{\mu}_{t-1}))^2}{2\sigma'^2}\right) \\
&= \mathcal{N}(\alpha_t x_t; y_t - (1-\alpha_t)\hat{\mu}_{t-1}, \sigma'^2)
\end{aligned}
\tag{49}
$$

We can now pull out the factor of $\alpha_t$ (again, for notational simplicity, we will write $\mu' = y_t - (1-\alpha_t)\hat{\mu}_{t-1}$):

$$
\begin{aligned}
\mathcal{N}(\star) &= \mathcal{N}(\alpha_t x_t; \mu', \sigma'^2) \\
&= \frac{1}{\sqrt{2\pi\sigma'^2}} \exp\left(-\frac{(\alpha_t x_t - \mu')^2}{2\sigma'^2}\right) \\
&= \frac{1}{\alpha_t\sqrt{2\pi\sigma'^2/\alpha^2}} \exp\left(-\frac{\alpha_t^2(x_t - \mu'/\alpha_t)^2}{2\sigma'^2}\right) \\
&= \frac{1}{\alpha_t\sqrt{2\pi\sigma''^2}} \exp\left(-\frac{(x_t - \mu'')^2}{2\sigma''^2}\right) \\
&= \frac{1}{\alpha_t}\mathcal{N}(x_t; \mu'', \sigma''^2)
\end{aligned}
\tag{50}
$$

where $\mu'' = \mu'/\alpha_t = (y_t - (1-\alpha_t)\hat{\mu}_{t-1})/\alpha_t$ and $\sigma''^2 = \sigma'^2/\alpha_t^2 = (1-\alpha_t)^2\hat{\sigma}_{t-1}^2/\alpha_t^2$. *Equation 49* can now be written as a Gaussian over $x_t$:

$$
\mathcal{N}(\star) = \frac{1}{\alpha_t}\mathcal{N}\left(x_t; (y_t - (1-\alpha_t)\hat{\mu}_{t-1})/\alpha_t, (1-\alpha_t)^2\hat{\sigma}_{t-1}^2/\alpha_t^2\right)
\tag{51}
$$

This allows us to combine the two distributions in *Equation 31*:

$$
\begin{aligned}
p\left(x_t | y_t, \hat{\theta}_{\tau<t}\right) &= \frac{1}{Z\left(y_t, \hat{\theta}_{\tau<t}\right)}\frac{1}{\alpha_t}\mathcal{N}(x_t; \mu_A, \sigma_A^2)\mathcal{N}(x_t; \mu_B, \sigma_B^2) \\
&= \frac{1}{Z\left(y_t, \hat{\theta}_{\tau<t}\right)}\frac{1}{\alpha_t}\frac{1}{\sqrt{2\pi(\sigma_A^2 + \sigma_B^2)}}\exp\left(-\frac{(\mu_A - \mu_B)^2}{2(\sigma_A^2 + \sigma_B^2)}\right)\mathcal{N}\left(x_t; \frac{\sigma_B^2\mu_A + \sigma_A^2\mu_B}{\sigma_A^2 + \sigma_B^2}, \frac{\sigma_A^2\sigma_B^2}{\sigma_A^2 + \sigma_B^2}\right) \\
&= \frac{f\left(y_t, \hat{\theta}_{\tau<t}\right)}{Z\left(y_t, \hat{\theta}_{\tau<t}\right)}\mathcal{N}\left(x_t; \frac{\sigma_B^2\mu_A + \sigma_A^2\mu_B}{\sigma_A^2 + \sigma_B^2}, \frac{\sigma_A^2\sigma_B^2}{\sigma_A^2 + \sigma_B^2}\right)
\end{aligned}
\tag{52}
$$

where:

$$
\begin{aligned}
\mu_A &= (y_t - (1-\alpha_t)\hat{\mu}_{t-1})/\alpha_t \\
\mu_B &= \hat{\mu}_{t-1} \\
\sigma_A^2 &= (1-\alpha_t)^2\hat{\sigma}_{t-1}^2/\alpha_t^2 \\
\sigma_B^2 &= \hat{\sigma}_{t-1}^2
\end{aligned}
\tag{53}
$$

Because the function $f\left(y_t, \hat{\theta}_{\tau<t}\right)$ does not depend on $x_t$, we can trivially obtain $Z\left(y_t, \hat{\theta}_{\tau<t}\right)$ by integrating over $x_t$ (as given by **Equation 7**):

$$
\begin{aligned}
Z\left(y_t, \hat{\theta}_{\tau<t}\right) &= \int dx_t\, p\left(y_t | x_t, \hat{\theta}_{\tau<t}\right) p\left(x_t | \hat{\theta}_{\tau<t}\right) \\
&= f\left(y_t, \hat{\theta}_{\tau<t}\right) \int dx_t\, \mathcal{N}\left(x_t; \frac{\sigma_B^2 \mu_A + \sigma_A^2 \mu_B}{\sigma_A^2 + \sigma_B^2}, \frac{\sigma_A^2 \sigma_B^2}{\sigma_A^2 + \sigma_B^2}\right) \\
&= f\left(y_t, \hat{\theta}_{\tau<t}\right)
\end{aligned}
\tag{54}
$$

The remaining terms in **Equation 52** are given by:

$$
\begin{aligned}
\sigma_A^2 + \sigma_B^2 &= \left(\frac{1 - 2\alpha_t + 2\alpha_t^2}{\alpha_t^2}\right) \hat{\sigma}_{t-1}^2 \\
\sigma_A^2 \sigma_B^2 &= \left(\frac{(1-\alpha_t)^2}{\alpha_t^2}\right) \hat{\sigma}_{t-1}^4 \\
\frac{\sigma_A^2 \sigma_B^2}{\sigma_A^2 + \sigma_B^2} &= \left(\frac{(1-\alpha_t)^2}{1 - 2\alpha_t + 2\alpha_t^2}\right) \hat{\sigma}_{t-1}^2 \\
\sigma_B^2 \mu_A + \sigma_A^2 \mu_B &= \left(\frac{\alpha_t y_t - (1-\alpha_t)(2\alpha_t - 1)\hat{\mu}_{t-1}}{\alpha_t^2}\right) \hat{\sigma}_{t-1}^2 \\
\frac{\sigma_B^2 \mu_A + \sigma_A^2 \mu_B}{\sigma_A^2 + \sigma_B^2} &= \frac{\alpha_t y_t - (1-\alpha_t)(2\alpha_t - 1)\hat{\mu}_{t-1}}{1 - 2\alpha_t + 2\alpha_t^2}
\end{aligned}
\tag{55}
$$

Putting everything together, the final form of **Equation 31** becomes:

$$
p\left(x_t | y_t, \hat{\theta}_{\tau<t}\right) = \mathcal{N}\left(x_t; \frac{\alpha_t y_t - (1-\alpha_t)(2\alpha_t - 1)\hat{\mu}_{t-1}}{1 - 2\alpha_t + 2\alpha_t^2}, \left(\frac{(1-\alpha_t)^2}{1 - 2\alpha_t + 2\alpha_t^2}\right) \hat{\sigma}_{t-1}^2\right)
\tag{56}
$$

For $\frac{1}{2} \le \alpha_t \le 1$, we can see that: $0 \le (1-\alpha_t)(2\alpha_t - 1) \le \frac{1}{8}$ and $\frac{1}{2} \le \left(1 - 2\alpha_t + 2\alpha_t^2\right) \le 1$.

# Appendix 2

DOI: https://doi.org/10.7554/eLife.32055.018

Here we provide a detailed derivation of the entropy of the output of filtering encoder (described in Section B.5.3).

To compute $H\left(y_t, y_{t+1}|\hat{\theta}_{\tau<t}\right)$, we assume that the encoder has access to the history of estimates $\hat{\theta}_{\tau<t}$, and that it uses the most recent estimate $\hat{\theta}_{t-1}$ as an approximate prediction of future states (i.e., $\hat{\theta}_{t-1} \approx \vec{\theta}_t \approx \vec{\theta}_{t+1}$).

For reference, the entropy of a normal distribution is:

$$H[\mathcal{N}(x;\mu,\sigma^2)] = \frac{1}{2}\log(2\pi e \sigma^2) \tag{57}$$

We want to compute $H\left(y_t, y_{t+1}|\hat{\theta}_{\tau<t}\right)$:

$$H\left(y_t, y_{t+1}|\hat{\theta}_{\tau<t}\right) = H\left(y_{t+1}|y_t, \hat{\theta}_{\tau<t}\right) + H\left(y_t|\hat{\theta}_{\tau<t}\right) \tag{58}$$

where $y_t = \alpha_t x_t + (1-\alpha_t)x_{t-1}$ is the output of the encoder, and $\alpha_t \in [0.5, 1]$ is the filtering coefficient.

To compute each of the terms in **Equation 58**, we need to compute $p\left(y_t|\hat{\theta}_{\tau<t}\right)$ and $p\left(y_{t+1}|y_t, \hat{\theta}_{\tau<t}\right)$. The first of these distributions is given by:

$$p\left(y_t|\hat{\theta}_{\tau<t}\right) = \mathcal{N}\left(y_t; \alpha_t\hat{\mu}_{t-1} + (1-\alpha_t)\hat{\mu}_{t-1}, \alpha_t^2\hat{\sigma}_{t-1}^2 + (1-\alpha_t)^2\hat{\sigma}_{t-1}^2\right), \tag{59}$$

whose entropy is given by:

$$H\left(y_t|\hat{\theta}_{\tau<t}\right) = \frac{1}{2}\log\left(2\pi e\left(\alpha_t^2\hat{\sigma}_{t-1}^2 + (1-\alpha_t)^2\hat{\sigma}_{t-1}^2\right)\right). \tag{60}$$

The second of these distributions can be written as:

$$p\left(y_{t+1}|y_t, \hat{\theta}_{\tau<t}\right) = \int dx_t\, p\left(y_{t+1}|x_t, \hat{\theta}_{\tau<t}\right) p\left(x_t|y_t, \hat{\theta}_{\tau<t}\right) \tag{61}$$

Noting that $p\left(y_{t+1}|x_{t+1}, x_t, \hat{\theta}_{\tau<t}\right) = \delta(y_{t+1} - (\alpha_t x_{t+1} + (1-\alpha_t)x_t))$, the first term in the integral in **Equation 61** is given by:

$$\begin{aligned}
p\left(y_{t+1}|x_t, \hat{\theta}_{\tau<t}\right) &= \int dx_{t+1}\, \delta(y_{t+1} - (\alpha_t x_{t+1} + (1-\alpha_t)x_t))\mathcal{N}\left(x_{t+1}; \hat{\mu}_{t-1}, \hat{\sigma}_{t-1}^2\right)\\
&= \frac{1}{\alpha_t}\mathcal{N}\left(\frac{(y_{t+1} - (1-\alpha_t)x_t)}{\alpha_t}; \hat{\mu}_{t-1}, \hat{\sigma}_{t-1}^2\right)\\
&= \frac{1}{(1-\alpha_t)}\mathcal{N}\left(x_t; \frac{(y_{t+1} - \alpha_t\hat{\mu}_{t-1})}{(1-\alpha_t)}, \frac{\alpha_t^2}{(1-\alpha_t)^2}\hat{\sigma}_{t-1}^2\right)
\end{aligned} \tag{62}$$

The second term in the integral in **Equation 61** is given by:

$$p\left(x_t|y_t, \hat{\theta}_{\tau<t}\right) = \mathcal{N}\left(x_t; \frac{\alpha_t y_t - (1-\alpha_t)(2\alpha_t - 1)\hat{\mu}_{t-1}}{\alpha_t^2 + (1-\alpha_t)^2}, \left(\frac{(1-\alpha_t)^2}{\alpha_t^2 + (1-\alpha_t)^2}\right)\hat{\sigma}_{t-1}^2\right) \tag{63}$$

Combining the two terms, we have:

$$p\left(y_{t+1}|x_t,\hat{\theta}_{\tau<t}\right)p\left(x_t|y_t,\hat{\theta}_{\tau<t}\right) = \frac{1}{(1-\alpha_t)}\mathcal{N}\left(x_t;\mu_A,\sigma_A^2\right)\mathcal{N}\left(x_t;\mu_B,\sigma_B^2\right)$$

$$= \frac{1}{(1-\alpha_t)}\frac{1}{\sqrt{2\pi(\sigma_A^2+\sigma_B^2)}}\exp\left(-\frac{(\mu_A-\mu_B)^2}{2(\sigma_A^2+\sigma_B^2)}\right) \quad (64)$$

$$\mathcal{N}\left(x_t;\frac{\sigma_B^2\mu_A+\sigma_A^2\mu_B}{\sigma_A^2+\sigma_B^2},\frac{\sigma_A^2\sigma_B^2}{\sigma_A^2+\sigma_B^2}\right)$$

where

$$\begin{aligned}
\mu_A &= \frac{(y_{t+1}-\alpha_t\hat{\mu}_{t-1})}{(1-\alpha_t)} \\
\mu_B &= \frac{\alpha_t y_t - (1-\alpha_t)(2\alpha_t-1)\hat{\mu}_{t-1}}{\alpha_t^2+(1-\alpha_t)^2} \\
\sigma_A^2 &= \frac{\alpha_t^2}{(1-\alpha_t)^2}\hat{\sigma}_{t-1}^2 \\
\sigma_B^2 &= \frac{(1-\alpha_t)^2}{\alpha_t^2+(1-\alpha_t)^2}\hat{\sigma}_{t-1}^2
\end{aligned} \quad (65)$$

Putting these terms back into the integral in **Equation 61** gives:

$$\begin{aligned}
p\left(y_{t+1}|y_t,\hat{\theta}_{\tau<t}\right) &= \frac{1}{(1-\alpha_t)}\frac{1}{\sqrt{2\pi(\sigma_A^2+\sigma_B^2)}}\exp\left(-\frac{(\mu_A-\mu_B)^2}{2(\sigma_A^2+\sigma_B^2)}\right) \\
&= \mathcal{N}\left(y_{t+1};\alpha_t\hat{\mu}_{t-1}+\frac{(1-\alpha_t)(\alpha_t y_t-(1-\alpha_t)(2\alpha-1)\hat{\mu}_{t-1})}{1-2\alpha+2\alpha_t^2},\alpha_t^2\hat{\sigma}_{t-1}^2+\frac{(1-\alpha_t)^4}{1-2\alpha_t+2\alpha_t^2}\hat{\sigma}_{t-1}^2\right)
\end{aligned} \quad (66)$$

The conditional entropy $H\left(y_{t+1}|y_t,\hat{\theta}_{\tau<t}\right)$ is determined by the variance in this distribution:

$$H\left(y_{t+1}|y_t,\hat{\theta}_{\tau<t}\right) = \frac{1}{2}\log\left(2\pi e\left(\alpha^2\hat{\sigma}_{t-1}^2+\frac{(1-\alpha)^4}{1-2\alpha+2\alpha^2}\hat{\sigma}_{t-1}^2\right)\right) \quad (67)$$

Combining the two entropy terms in **Equations 60 and 67**, we get:

$$\begin{aligned}
H\left(y_t,y_{t+1}|\hat{\theta}_{\tau<t}\right) &= H\left(y_{t+1}|y_t,\hat{\theta}_{\tau<t}\right)+H\left(y_t|\hat{\theta}_{\tau<t}\right) \\
&= \frac{1}{2}\log\left(4\pi^2 e^2\left(\alpha_t^2\hat{\sigma}_{t-1}^2+\frac{(1-\alpha_t)^4}{1-2\alpha_t+2\alpha_t^2}\hat{\sigma}_{t-1}^2\right)\left(\alpha_t^2\hat{\sigma}_{t-1}^2+(1-\alpha_t)^2\hat{\sigma}_{t-1}^2\right)\right)
\end{aligned} \quad (68)$$

