## [Decision Letter]

Thank you for submitting your article "Adaptive coding for dynamic sensory inference" for consideration by *eLife*. Your article has been reviewed by three peer reviewers, one of whom is a member of our Board of Reviewing Editors, and the evaluation has been overseen by Timothy Behrens as the Senior Editor. The reviewers have opted to remain anonymous.

The reviewers have discussed the reviews with one another and the Reviewing Editor has drafted this decision to help you prepare a revised submission.

Summary:

The manuscript proposes a modeling framework to unify efficient coding of sensory stimuli with Bayesian inference of a latent, behaviorally-relevant variable (e.g. an environmental state that generates the sensory stimulus). The main idea is to replace the traditional objective for efficient coding – quality of the sensory stimulus reconstruction – with an "inference cost", namely the discrepancy between the estimate of the latent state based on the neural code and the estimate of the Bayesian ideal observer. Optimizing for these two objectives, under the constraint of limited metabolic resources, gives qualitatively different results. In particular, optimizing for inference cost produces encoding schemes that are less metabolically expensive and more accurate (with respect to the latent state).

Overall, the reviewers found this to be a very nice contribution towards a unifying framework for these ideas. The manuscript states clearly and accurately addresses this research question. The figures are well laid-out, visually appealing, and informative.

However, several revisions are requested to 1) show a relation to biological data, 2) demonstrate the generality of the results, 3) more clearly state what areas of the brain this work applies to, and 4) improve clarity of the presentation by reducing the overall length of the article.

Essential revisions:

1) The work should include more explicit relation to biological data, both published results and the authors' predictions from their work on what one might expect to see in data. Please provide some examples that link predictions of the proposed framework to published data.

Example: when discussing metamers, the authors state that "stimuli become less distinguishable to the observer as its model of the environment becomes more accurate". How does this relate to the empirical observation that visual metamers are predominant in the periphery of the visual field where resolution is low and presumably the model less accurate, but can be resolved by foveal inspection where resolution is higher and the model more accurate?

Another example: could the predictions of Figure 6 and the final paragraph of subsection “Limited gain and temporal acuity” be related to data on the stimulus-dependence of adaptation in the visual cortex?

2) The manuscript needs to address the generality of these results. A very simple stimulus model with switching between two states was used; how do these results extend to more complex stimuli? Does the work here predict that these results are general and, if so, for which stimulus conditions? Substantive text revisions and additional computational results will be needed to satisfy this point.

The authors discuss the dynamical signatures that distinguish the three encoding schemes (Figure 9). To make a higher impact, they should also provide or suggest examples from biology where one of the three schemes may be more or less likely. Where would one expect to see mean vs variance changing environments? What would one have to measure to observe the dynamical signatures in Figure 9? Can the authors make some predictions/hypotheses about this?

In addition, in the context of the experimental system where the three schemes might apply, how likely is it that one would be able to distinguish between each scheme? In the panels in Figure 9, some of the differences between the different schemes are very small so they might not be measurable with great precision in specific experiments.

In particular, the reviewers noted that actual sensory stimuli are high-dimensional, and behaviorally-relevant latent states (and behavioral output itself) are low-dimensional. The fact that the code optimized for inference leads to better accuracy and efficiency, compared to the code optimized for reconstruction, is true for the latent variable that has been deemed behaviorally relevant, but the result would probably be different in a more realistic generative model in which there are also other latent variables that jointly generate high-d stimuli. Also, as we know from natural image and sound statistics, stimulus distributions are not well described by Gaussians or even mixtures of Gaussians. Related to this, the definition of inference cost as the expected squared error makes sense if assuming Gaussian posteriors, but perhaps the authors should use something more general to encompass a broader range of cases, like D_KL (understanding that's effectively what the authors have done for the Gaussian case).

In the Discussion, the authors claim they have "addressed the issues of both tractability and non-stationarity…", but it is not clear if this is because of the simplifying assumptions that were made for the generative model of stimuli and for the inference cost.

3) It was not clear to the reviewers if this is a theory that applies to the sensory periphery, to the cortex, or to both. The resource limitations considered here make sense if the dimensionality of the stimulus x and neural code y are the same. That would seem to be appropriate for a theory of the periphery, but then optimizing a peripheral code in complex ways for behavioral outputs may be implausible. Conversely, if this applies to cortex, we know dim(y)>>dim(x) and the choice of resource limitations might not be as relevant in practice: e.g. wouldn't a large enough population code overcome the problem of discrete response levels? It has been argued that in cortex the major challenge is the computational complexity of the inferences that need to be performed (Beck et al., 'not noisy just wrong'), and that approximate inference may be more a important constraint than resource limitations.

4) The paper would benefit from some streamlining to reduce the number of figures and the overall length of the paper. Repetitive text should be condensed or eliminated (example: Results section, fourth paragraph is a repeat of earlier statements). Overall, the Introduction could be significantly condensed. It is suggested that Box 1 be moved to the Materials and methods section, because there is significant overlap with it and Figure 1. The figures could be streamlined to some extent, perhaps a few could be combined to reduce the total figure count. At times that the manuscript was hard to read, as long paragraphs are spent describing the mechanics of the effects (which are instead very clearly illustrated in the figures). The authors should consider shortening those descriptions.

[Editors' note: further revisions were requested prior to acceptance, as described below.]

Thank you for resubmitting your work entitled "Adaptive coding for dynamic sensory inference" for further consideration at *eLife*. Your revised article has been favorably evaluated by Timothy Behrens (Senior editor), a Reviewing editor, and two reviewers.

Thank you for your extensive and substantive revision of your paper. The manuscript has been improved, and reviewers were largely satisfied with your changes, but there are some remaining issues that need to be addressed before acceptance, as outlined below:

1) On the question of resource limitations, the reviewers agree that your argument re: sparse coding is valid, but it does raise questions about noise robustness (connected to point 2 below) and cell death, which might both argue for distributed/redundant coding. Perhaps discussing/citing Deneve's recent work on degenerate population codes is appropriate here. Please add some text to the discussion that addresses this.

2) Regarding adding noise to your model: Please include a discussion of the impact of noise on the structure of neural coding along with your text clearly stating that you have left the work of a thorough exploration of these topics for another study. Relevant literature to discuss/cite are, e.g. Zohary et al., 1994; Abbott Dayan, 1999; Sompolinsky et al., 2001; Ma et al., 2006; Moreno-Bote et al., 2014; a review article by Kohn et al., 2016; as well as Gjorgjieva et al. bioRxiv 2017 (currently cited in the wrong place – should be moved to the discussion of the first framework). Please add some text to the Discussion that addresses this. If possible, the reviewers urge the authors to include one example of the effects of noise (a simple additive Gaussian or Poisson noise source) on their model results.

Points 1 and 2 are connected and that should also be made clear in the added Discussion text.

3) Please address in the Discussion how your results might be thought of in the context of dynamic environments that are stationary – i.e. the stimulus changes in time, and might *also* switch states, but any given state fluctuates.

---

## [Author Response]

Essential revisions:1) The work should include more explicit relation to biological data, both published results and the authors' predictions from their work on what one might expect to see in data. Please provide some examples that link predictions of the proposed framework to published data.Example: when discussing metamers, the authors state that "stimuli become less distinguishable to the observer as its model of the environment becomes more accurate". How does this relate to the empirical observation that visual metamers are predominant in the periphery of the visual field where resolution is low and presumably the model less accurate, but can be resolved by foveal inspection where resolution is higher and the model more accurate?Another example: could the predictions of Figure 6 and the final paragraph of subsection “Limited gain and temporal acuity” be related to data on the stimulus-dependence of adaptation in the visual cortex?

First, we have added a new analysis that extends our adaptive coding framework to a more naturalistic inference task. We model this task after computations that are known to occur in the visual pathway. This includes a sparse coding model that mimics receptive fields in V1, and a projection onto curvature filters that mimics computations in V2. The output of V2 filters is used to adapt the sparsity of the population code in V1. We use this model to infer changes in local curvature of a natural image when gaze shifts from one region of the image to another. We show that this model exhibit bursts of population activity when stimuli (local image patches) are surprising, or when the observer is uncertain, consistent with the general principles that we now use to frame the paper. We link this result to a recent study that finds bursts of activity in V1 in response to stimuli that violate statistical regularities in the environment. These results can be found in a new section entitled “Adaptive coding for inference under natural conditions”, and the corresponding figure (now Figure 7).

Second, we have clarified two key signatures of encoding schemes optimized for inference: bursts to signal salient changes in the environment, and ambiguous stimulus representations when the environment is stationary. We cite a broad range of published work that provides evidence for both of these dynamical signatures; this work spans several different sensory modalities and different stages of processing. Where available, we provide evidence that these dynamics are modulated by predictive feedback from downstream areas (consistent with the feedback projections that we use to adapt the encoding scheme) and are relevant for behavior. These citations can be found in two sections of the Discussion entitled “Transient increases in fidelity signal salient changes in the environment” and “Periods of stationarity give rise to ambiguous stimulus representations”.

In response to this and other comments, we have revised our discussion of metamers. We now remove the term metamer in favor a description of the effect, namely that in stationary environments, physically different stimuli will be increasingly likely to be perceived as similar as the observer’s model becomes aligned with the environment. We think that this phenomenon, in which the discriminability of stimuli decreases over time, is consistent with the observation of auditory metamers (as discussed in the Discussion section entitled “Periods of stationarity give rise to ambiguous stimulus representations”). It might be possible to extend our framework to the study of visual metamers, as the reviewers propose. Here, the notion of “accuracy” that the reviewers mention is related to the resolution of receptive fields in the fovea relative to the periphery. This is similar to the high- and low-resolution response levels in our discretization scheme, which we propose should be dynamically shifted over time to improve the accuracy of the observer’s model (which is distinct from the resolution of response levels used to construct this model). We think that the scenario of visual metamers could map more naturally onto an active sensing scheme in which the visual system can shift its high foveal resolution to different parts of a visual scene in order to extract information about the underlying spatiotemporal statistics. Active sensing is beyond the scope of the current paper, and as such, we have chosen not to elaborate on this example. We do, however, think that this is an interesting direction for future work.

2) The manuscript needs to address the generality of these results. A very simple stimulus model with switching between two states was used; how do these results extend to more complex stimuli? Does the work here predict that these results are general and, if so, for which stimulus conditions? Substantive text revisions and additional computational results will be needed to satisfy this point.

We have added two new sets of analyses (now shown in Figure 1 and Figure 7) and have made substantive changes to the text in order to address the generality of our results. First, we have identified and clearly stated two important principles that shape efficient coding for inference: 1) the relative utility of incoming stimuli for inference can change over time, and 2) physically different stimuli can exert a similar influence on the observer’s model of the environment, and can therefore be encoded in the same neural response without affecting the inference process. We show that both principles are shaped by uncertainty in the observer’s belief about the state of the environment, and by the surprise of incoming stimuli given this belief (Figure 1B-C). We then show that the qualitative features of this relationship between surprise, uncertainty, and the dynamics of inference hold for the estimation of both location (analogous to mean) and scale (analogous to variance) of a generalized Gaussian distribution (Figure 1D). The parameters of the generalized Gaussian distribution can be varied to generate many specific distributions (including Laplace, Gaussian, and flat), and as such can capture statistical properties of natural stimuli (see, e.g., [1]). Finally, we show that this qualitative relationship is observed in a more realistic scenario using natural image stimuli and modeled after computations in the visual pathway (Figure 7).

We note, however, that the detailed geometry of the relationship between surprise, uncertainty, and inference can change depending on the specific model of the environment. Developing a full Bayesian observer model for more naturalistic stimuli is an interesting direction for future work, but we anticipate that such a model will rely on surprise and uncertainty in a manner that is qualitatively similar to the systems explored here.

The authors discuss the dynamical signatures that distinguish the three encoding schemes (Figure 9). To make a higher impact, they should also provide or suggest examples from biology where one of the three schemes may be more or less likely. Where would one expect to see mean vs variance changing environments? What would one have to measure to observe the dynamical signatures in Figure 9? Can the authors make some predictions/hypotheses about this?In addition, in the context of the experimental system where the three schemes might apply, how likely is it that one would be able to distinguish between each scheme? In the panels in Figure 9, some of the differences between the different schemes are very small so they might not be measurable with great precision in specific experiments.

In response to these and other comments, we have revised our claims about the dynamical signatures of these encoding strategies. We agree that some of the differences between encoding schemes are very small. We now highlight the fact that all three encoding schemes produce qualitatively similar response properties when optimized for inference, and these response properties differ from those observed when the same encoding schemes are optimized for stimulus reconstruction. We highlight these similarities and differences in a new figure (Figure 6), and we have significantly revised the accompanying text (which can be found in the Results section entitled “Dynamical signatures of adaptive coding”). In the Discussion, we have added numerous examples of where similar dynamical signatures have been observed experimentally. These examples span both physiology and behavior, and they encompass many different sensory modalities, including vision, audition, and olfaction (see the sections entitled “Transient increases in fidelity signal salient changes in the environment” and “Periods of stationarity give rise to ambiguous stimulus representations”). Several of these studies use simple distributions of stimuli in which the mean or the variance of the stimulus distribution is switching over time. We specifically highlight examples of these types of stimulus environments, and we discuss the utility of these environments for studying inference.

However, as before, we do observe that the three encoding schemes can impact the inference process in qualitatively different ways. Rather than demonstrating this with a visual comparison between inference trajectories (as was previously shown in Figure 9), we quantify these differences in a set of new figure panels, Figure 6D-E (which also addresses a later comment). We then show that the asymmetry in the speed of inference for upward versus downward switches in variance takes a qualitatively different form for each encoding scheme: one encoding scheme accentuates this asymmetry, another nearly removes it, and a third reverses it (as compared to the optimal Bayesian model in the absence of an encoding). We believe that this difference in the relative speed of responses to upward versus downward switches provides a stronger test of the underlying encoding scheme, without needing to rely on small quantitative differences. These findings are discussed in the Results section entitled “Dynamical signatures of adaptive coding”.

We feel that it would be a misrepresentation of this work to claim that individual encoding schemes are particular to certain brain regions or stages of neural processing. Rather, we view each encoding scheme as a simplification of a particular neural computation, which can be implemented in different parts of the nervous system. When introducing each encoding scheme, we provide examples of other studies that have used similar models to describe neural computations. In the Discussion, we now highlight the features of each encoding scheme, and we hypothesize conditions under which each scheme might be useful.

In particular, the reviewers noted that actual sensory stimuli are high-dimensional, and behaviorally-relevant latent states (and behavioral output itself) are low-dimensional. The fact that the code optimized for inference leads to better accuracy and efficiency, compared to the code optimized for reconstruction, is true for the latent variable that has been deemed behaviorally relevant, but the result would probably be different in a more realistic generative model in which there are also other latent variables that jointly generate high-d stimuli.

By construction, we expect a code optimized for inference to yield better inference accuracy than a code optimized for reconstruction (regardless of the dimensionality of the latent space). We now clarify this in the:

“As expected, a strategy optimized for inference achieves lower inference error than a strategy optimized for stimulus reconstruction (across all numbers of response levels), but it also does so at significantly lower coding cost.”

We agree that in a more realistic scenario in which the latent space is higher dimensional, the cost of encoding for inference could increase. However, we argue that even in complex latent spaces, an encoding scheme optimized for inference should adapt based on uncertainty and surprise and will therefore exhibit qualitatively different features than an encoding scheme optimized for reconstruction. We now highlight this in the Discussion:

“In such cases, we expect the dimensionality of the latent variable space to determine the lower bound on coding costs for inference. Even in the limit of highly complex models, however, we expect accurate inference and reconstruction to impose qualitatively different constraints on neural response properties.”

Also, as we know from natural image and sound statistics, stimulus distributions are not well described by Gaussians or even mixtures of Gaussians. Related to this, the definition of inference cost as the expected squared error makes sense if assuming Gaussian posteriors, but perhaps the authors should use something more general to encompass a broader range of cases, like D_KL (understanding that's effectively what the authors have done for the Gaussian case).

As described above, we have shown that the qualitative relationship between surprise, uncertainty, and inference dynamics extends beyond a Gaussian distribution. In particular, we demonstrate that the same qualitative relationship holds for the estimation of the location and scale of a generalized Gaussian distribution with a range of different parameters (corresponding to Laplace, Gaussian, and flat distributions). We have also shown that these principles apply to a more naturalistic inference scenario using natural image patches.

We also stress the difference between the stimulus distribution (*p(x_t_|θ_t_*)) and the prior and posterior distributions over parameter values (*p(θ_t_|y_τ<t_*) and *p(θ_t_|y_τ≤t_)*, respectively). While in our simulations, the stimulus distribution is indeed a mixture of Gaussians, the majority of the prior and posterior distributions considered here are non-Gaussian: the posterior distributions used in the mean- and variance-switching environments are bimodal, and the prior and posterior distributions over scale parameters in Figure 1 are inverse Γ functions. Moreover, the choice of mean squared error (MSE) as a measure of inference cost does not make any assumptions about the shape of the posterior. MSE is a cost function that is guaranteed to be minimized by the mean of the posterior distribution, regardless of the form of the posterior [2, 3]. In this sense, MSE is a fully general cost function and does not reflect any particular assumptions about Gaussianity. We now motivate our use of MSE as a measure of inference cost, and we highlight its generality:

“In order to optimize and assess the dynamics of the system, we use the point values θ^t and θ→*_t+1_* as an estimate of the current state and prediction of the future state, respectively. The optimal point estimate is computed by averaging the posterior and is guaranteed to minimize the mean squared error between the estimated state θ^t and the true state *θ_t_*, regardless of the form of the posterior distribution.”

As mentioned by the reviewer, there are other measures of inference cost, including the KL-divergence of the posterior from the prior distribution. Such a measure would take into account not only the difference in the mean of the posterior, but also a change of uncertainty after incorporating a new stimulus sample. We have noted this explicitly in the Discussion, and we agree that this is an interesting generalization to be explored in future work:

“Other measures, such as KL-divergence, could be used to capture not only changes in point estimates, but also changes in uncertainty underlying these estimates.”

In the Discussion, the authors claim they have "addressed the issues of both tractability and non-stationarity…", but it is not clear if this is because of the simplifying assumptions that were made for the generative model of stimuli and for the inference cost.

As described above, our choice of inference cost does not make any assumptions about the form of the inference model. Moreover, we have demonstrated that surprise and uncertainty shape inference across a range of different models of stimulus generation. Nevertheless, we have revised this sentence in the Discussion so as not to overstate our claims:

“Here, we frame general principles that constrain the dynamic balance between coding cost and task relevance, and we pose neurally-plausible implementations.”

3) It was not clear to the reviewers if this is a theory that applies to the sensory periphery, to the cortex, or to both. The resource limitations considered here make sense if the dimensionality of the stimulus x and neural code y are the same. That would seem to be appropriate for a theory of the periphery, but then optimizing a peripheral code in complex ways for behavioral outputs may be implausible. Conversely, if this applies to cortex, we know dim(y)>>dim(x) and the choice of resource limitations might not be as relevant in practice: e.g. wouldn't a large enough population code overcome the problem of discrete response levels? It has been argued that in cortex the major challenge is the computational complexity of the inferences that need to be performed (Beck et al., 'not noisy just wrong'), and that approximate inference may be more a important constraint than resource limitations.

We postulate that the principles discussed in this paper can bear relevance for the entire sensory hierarchy, from periphery to central areas. We believe this to be the case based on two observations. First, neurons in all brain regions perform computations that operate on input from upstream areas, and these computations can frequently be described as probabilistic inference (e.g. [4, 5]). Second, energy limitations shape neuronal communication across the nervous system [6]. Our framework specifies how to bridge these two widespread phenomena. We now address this directly at the beginning of the Discussion.

The fact that cortex is high dimensional does not mean that resource limitations are irrelevant; one could alternatively argue that efficient energy use (at the single neuron level) becomes even more important as the system increases in size. The adaptive coding schemes discussed in this paper could be applied at the single neuron level, or they could be formulated for population codes. In either case, the number of neurons in a population places an upper bound on energy expenditure. By appropriately adapting its neural code, however, the system can operate well below this limit. In fact, it has been argued that the sparse activity observed in cortex (particularly during natural stimulation) is a demonstration of this type of efficiency [7, 8, 6].

While the majority of the paper focuses on cases where the dimensionality of the stimulus and the neural code are the same, the general principles of this framework apply to stimuli and representations of arbitrary dimensionality. We address this issue by simulating a population of model neurons responding to natural image patches (Figure 7). For simplicity, we chose the dimensionality of the neural response to be lower than the dimensionality of the stimulus; however, we expect the qualitative features of these results to hold for scenarios in which the dimensionality of the neural response is larger than the dimensionality of the stimulus. This is because the observer’s belief is most strongly affected by the surprise of incoming stimuli during periods of uncertainty, regardless of the dimensionality of the neural population in which these stimuli are encoded.

*4) The paper would benefit from some streamlining to reduce the number of figures and the overall length of the paper. Repetitive text should be condensed or eliminated (example: Results section, fourth paragraph is a repeat of earlier statements). Overall, the Introduction could be significantly condensed. It is suggested that Box 1 be moved to the Materials and methods section, because there is significant overlap with it and Figure 1. The figures could be streamlined to some extent, perhaps a few could be combined to reduce the total figure count. At times that the manuscript was hard to read, as long paragraphs are spent describing the mechanics of the effects (which are instead very clearly illustrated in the figures). The authors should consider shortening those descriptions.*

We have reduced the original set of 9 figures to 6 figures (Figure 1-6), we have added one additional Figure (Figure 7) to help address concerns 1-3 above, and we have moved Box 1 to the beginning of the Materials and methods section (now labeled “Figure 1—figure supplement 1”). This has reduced the number of graphical elements in the main text from 10 to 7. We have additionally streamlined the text by removing redundant statements, shortening descriptions of the mechanics of the effects, and condensing the Introduction. This streamlining was done throughout the text, so we do not enumerate the changes here, but we have marked all changes to the text in a separate document.

[Editors' note: further revisions were requested prior to acceptance, as described below.]

1) On the question of resource limitations, the reviewers agree that your argument re: sparse coding is valid, but it does raise questions about noise robustness (connected to point 2 below) and cell death, which might both argue for distributed/redundant coding. Perhaps discussing/citing Deneve's recent work on degenerate population codes is appropriate here. Please add some text to the discussion that addresses this.2) Regarding adding noise to your model: Please include a discussion of the impact of noise on the structure of neural coding along with your text clearly stating that you have left the work of a thorough exploration of these topics for another study. Relevant literature to discuss/cite are, e.g. Zohary et al. 1994; Abbott Dayan 1999; Sompolinsky et al., 2001; Ma et al., 2006; Moreno-Bote et al., 2014; a review article by Kohn et al., 2016; as well as Gjorgjieva et al. bioRxiv 2017 (currently cited in the wrong place – should be moved to the discussion of the first framework). Please add some text to the Discussion that addresses this. If possible, the reviewers urge the authors to include one example of the effects of noise (a simple additive Gaussian or Poisson noise source) on their model results.Points 1 and 2 are connected and that should also be made clear in the added Discussion text.

We have revised the Discussion to include a section dedicated to the discussion of noise robustness. Within this section, we have cited a broad body of literature (including references highlighted by the reviewers) that addresses the role of noise on the structure of the neural code. While we agree that cell death is an interesting source of potential fragility within neural population codes, we feel that this is outside the scope of the present study. We have nevertheless cited Deneve’s recent work on degenerate population codes in the context of noise robustness.

As requested, we have run additional simulations to demonstrate the effects of additive Gaussian noise on our results. These results are shown in Figure 3—figure supplement 2, and are highlighted in the Discussion. We find that the accuracy of the inference process is robust to low levels of noise, but degrades significantly when noise levels approach the separation between latent states in the environment.

We acknowledge directly in the text that this example is one of many potential sources of noise, each of which can have differing effects on the structure of optimal codes. As mentioned by the reviewers, a thorough investigation of these issues is the subject of future work, and we now state this directly in the text.

These changes can be found in the following paragraphs:

“Noise can arise at different stages of neural processing, and can alter the faithful encoding and transmission of stimuli to downstream areas [9, 10]. Individual neurons and neural populations can combat the adverse effects of noise by appropriately tuning their coding strategies, for example by adjusting the gain or thresholds of individual neurons [11, 12], introducing redundancies between neural responses [13,14, 15, 16, 17], and forming highly distributed codes [18, 19]. Such optimal coding strategies depend on the source, strength, and structure of noise [10, 14, 11, 20], and can differ significantly from strategies optimized in the absence of noise [13].

“Noise induced during encoding stages can affect downstream computations, such as the class of inference tasks considered here. To examine its impact on optimal inference, we injected additive Gaussian noise into the neural response transmitted from the encoder to the observer. We found that the accuracy of inference was robust to low levels of noise, but degraded quickly once the noise variance approached the degree of separation between environmental states (Figure 3—figure supplement 2). Although this form of Gaussian transmission noise was detrimental to the inference process, previous work has argued that noise-related variability, if structured appropriately across a population of encoders, could support representations of the probability distributions required for optimal inference [21]. Moreover, we expect that the lossy encoding schemes developed here could be beneficial in combating noise injected prior to the encoding step, as they can guarantee that metabolic resources are not wasted in the process of representing noise fluctuations.

“Ultimately, the source and degree of noise can impact both the goal of the system and the underlying coding strategies. Here, we considered the goal of optimally inferring changes in environmental states. However, in noisy environments where the separation between latent environmental states is low, a system might need to remain stable in the presence of noise, rather than flexible to environmental changes. We expect that the optimal balance between stability and flexibility to be modulated by the spread of the stimulus distribution relative to the separation between environmental states. A thorough investigation of potential sources of noise, and their impact on the balance between efficient coding and optimal inference, is the subject of future work.”

3) Please address in the Discussion how your results might be thought of in the context of dynamic environments that are stationary – i.e. the stimulus changes in time, and might *also* switch states, but any given state fluctuates.

We have revised our Discussion of our simple environment model to highlight the possibility that the environmental state could both fluctuate in time and switch states:

“Here, we used this simple environment to probe the dynamics of encoding schemes optimized for inference. We found that optimal encoding schemes respond strongly to changes in the underlying environmental state, and thereby carry information about the timescale of environmental fluctuations. In natural settings, signals vary over a range of temporal scales, and neurons are known to be capable of adapting to multiple timescales in their inputs. We therefore expect that more complex environments, for example those in which the environmental state can both switch between distinct distributions and fluctuate between values within a single distribution, will require that the encoder respond to environmental changes on multiple timescales.”

[1] Y. Karklin and M. S. Lewicki, “A hierarchical bayesian model for learning nonlinear statistical regularities in nonstationary natural signals,” *Neural computation*, vol. 17, no. 2, pp. 397–423, 2005.

[2] E. T. Jaynes, *Probability theory: The logic of science*. Cambridge university press, 2003.

[3] C. Robert, *The Bayesian choice: from decision-theoretic foundations to computational implementation*. Springer Science & Business Media, 2007.

[4] T. S. Lee and D. Mumford, “Hierarchical bayesian inference in the visual cortex,” *JOSA A*, vol. 20, no. 7, pp. 1434–1448, 2003.

[5] R. Coen-Cagli, A. Kohn, and O. Schwartz, “Flexible gating of contextual influences in natural vision,” *Nature neuroscience*, vol. 18, no. 11, p. 1648, 2015.

[6] P. Sterling and S. Laughlin, *Principles of neural design*. MIT Press, 2015.

[7] M. R. DeWeese and A. M. Zador, “Binary coding in auditory cortex,” in *Advances in neural information processing systems*, pp. 117–124, 2003.

[8] W. E. Vinje and J. L. Gallant, “Sparse coding and decorrelation in primary visual cortex during natural vision,” *Science*, vol. 287, no. 5456, pp. 1273–1276, 2000

[9] W. Bialek, F. Rieke, R. de Ruyter van Steveninck, and D. Warland, “Spikes: exploring the neural code,” *MIT. Roddey, JC, Girish, B., & Miller, JP (2000). Assessing the performance of neural encoding models in the presence of noise. Journal of Computational Neuroscience*, vol. 8, no. 95, p. 112, 1997.

[10] B. A. Brinkman, A. I. Weber, F. Rieke, and E. Shea-Brown, “How do efficient coding strategies depend on origins of noise in neural circuits?,” *PLoS computational biology*, vol. 12, no. 10, p. e1005150, 2016.

[11] J. Van Hateren, “A theory of maximizing sensory information,” *Biological Cybernetics*, vol. 68, no. 1, pp. 23–29, 1992.

[12] J. Gjorgjieva, M. Meister, and H. Sompolinsky, “Optimal sensory coding by populations of on and off neurons,” *bioRxiv*, p. 131946, 2017.

[13] E. Doi and M. S. Lewicki, “A simple model of optimal population coding for sensory systems,” *PLoS computational biology*, vol. 10, no. 8, p. e1003761, 2014.

[14] G. Tkačik, J. S. Prentice, V. Balasubramanian, and E. Schneidman, “Optimal population coding by noisy spiking neurons,” *Proceedings of the National Academy of Sciences*, vol. 107, no. 32, pp. 14419–14424, 2010.

[15] R. Moreno-Bote, J. Beck, I. Kanitscheider, X. Pitkow, P. Latham, and A. Pouget,

“Information-limiting correlations,” *Nature Neuroscience*, vol. 17, pp. 1410–1417, 2014.

[16] L. Abbott and P. Dayan, “The effect of correlated variability on the accuracy of a population code,” *Neural Computation*, vol. 11, no. 1, pp. 91–101, 1999.

[17] H. Sompolinsky, H. Yoon, K. Kang, and S. M, “Population coding in neuronal systems with correlated noise,” *Physical Review E*, vol. 64, p. 051904, 2001.

[18] S. Denève and C. Machens, “Efficient codes and balanced networks,” *Nature Neuroscience*, vol. 19, p. 375–382, 2016.

[19] S. Denève and M. Chalk, “Efficiency turns the table on neural encoding, decoding and noise,” *Current Opinion in Neurobiology*, vol. 37, pp. 141–148, 2016.

[20] A. Kohn, R. Coen-Cagli, I. Kanitscheider, and A. Pouget, “Correlations and neuronal population information,” *Annual Reviews of Neuroscience*, vol. 39, pp. 237–256, 2016.

[21] W. Ma, J. Beck, P. Latham, and A. Pouget, “Bayesian inference with probabilistic population codes,” *Nature Neuroscience*, vol. 9, p. 1432–1438, 2006.